# Mapping the aerodynamic roughness of the Greenland Ice Sheet surface using ICESat-2: Evaluation over the K-transect

Maurice van Tiggelen[1], Paul C. J. P. Smeets[1], Carleen H. Reijmer[1], Bert Wouters[1,3], Jakob F. Steiner[2,4], Emile J. Nieuwstraten[2], Walter W. Immerzeel[2], and Michiel R. van den Broeke[1]

[1]Institute for Marine and Atmospheric research (IMAU), Utrecht University, Utrecht, the Netherlands
[2]Department of Physical Geography, Utrecht University, Utrecht, the Netherlands
[3]Department of Geoscience and Remote Sensing, Delft University of Technology, Delft, the Netherlands
[4]International Centre for Integrated Mountain Development, Kathmandu, Nepal

**Correspondence:** Maurice van Tiggelen (m.vantiggelen@uu.nl)

**Abstract.** The aerodynamic roughness of heat, moisture and momentum of a natural surface is an important parameter in atmospheric models, as it co-determines the intensity of turbulent transfer between the atmosphere and the surface. Unfortunately this parameter is often poorly known, especially in remote areas where neither high-resolution elevation models nor eddy-covariance measurements are available. In this study we adapt a bulk drag partitioning model to estimate the aerodynamic roughness length ($z_{0m}$) such that it can be applied to 1D (i.e. unidirectional) elevation profiles, typically measured by laser altimeters. We apply the model to a rough ice surface on the K-transect (west Greenland Ice Sheet) using UAV photogrammetry, and evaluate the modelled roughness against in situ eddy-covariance observations. We then present a method to estimate the topography at 1 m horizontal resolution using the ICESat-2 satellite laser altimeter, and demonstrate the high precision of the satellite elevation profiles against UAV photogrammetry. The currently available satellite profiles are used to map the aerodynamic roughness during different time periods along the K-transect, that is compared to an extensive dataset of in situ observations. We find a considerable spatio-temporal variability in $z_{0m}$, ranging between $10^{-4}$ m for a smooth snow surface over $10^{-1}$ m for rough crevassed areas, which confirms the need to incorporate a variable aerodynamic roughness in atmospheric models over ice sheets.

## 1 Introduction

Between 1992 and 2018, the mass loss of the Greenland Ice Sheet (GrIS) contributed $10.8 \pm 0.9$ mm to global mean sea-level rise (Shepherd et al., 2020). This mass loss is caused in approximately equal parts by an increase in ice discharge (Mouginot et al., 2019; King et al., 2020), and an increase in surface meltwater runoff (Noël et al., 2019). Runoff occurs mostly in the low-lying ablation area of the GrIS, where bare ice is exposed to on-average positive air temperatures throughout summer (Smeets et al., 2018; Fausto et al., 2021). As a consequence, the downward turbulent mixing of warmer air towards the bare ice, the

sensible heat flux, is an important driver of GrIS mass loss next to radiative fluxes (Fausto et al., 2016; Kuipers Munneke et al., 2018; Van Tiggelen et al., 2020).

Although the strong vertical temperature gradient provides the required source of energy, it is the persistent katabatic winds that generate the turbulent mixing through wind shear (Forrer and Rotach, 1997; Heinemann, 1999). Additionally, the surface

of the GrIS close to the ice edge is very rough (Yi et al., 2005; Smeets and Van den Broeke, 2008). It is composed of closely spaced obstacles, such as ice hummocks, crevasses, melt streams and moulins. Due to the effect of form drag (or pressure drag) $\tau_r$, the magnitude of the turbulent fluxes increases with surface roughness (e.g. Garratt, 1992), thereby enhancing surface melt (Van den Broeke, 1996; Herzfeld et al., 2006). As of today, the effect of form drag on the sensible heat flux over the GrIS, and therefore its impact on surface runoff, remains poorly known.

The first challenge in modelling this turbulent mixing resides in accurately modelling the surface shear stress, without the need to calculate the detailed air pressure distribution around each individual surface obstacle. Such bulk drag models have been developed by e.g. Arya (1975) to estimate the drag caused by pressure ridges on Arctic pack ice. This model was extended by Hanssen-Bauer and Gjessing (1988) for varying sea-ice concentrations. A more general drag model was proposed by Raupach (1992), that was extended by Andreas (1995) for sastrugi, by Smeets et al. (1999) for rough ice and by Shao and Yang (2008)

for surfaces with higher obstacle density, such as urban areas. Lüpkes et al. (2012) and Lüpkes and Gryanik (2015) developed a bulk drag model for sea-ice that is used in multiple atmospheric models. Over glaciers, semi-empirical approaches based on Lettau (1969) are often used, such as by Munro (1989), Fitzpatrick et al. (2018) and Chambers et al. (2019).

The second challenge is the application of such models in weather and climate models, which requires mapping small-scale obstacles over large areas, e.g. an entire glacier or ice sheet. Historically, the surveying of rough ice was spatially limited to

areas accessible for instrument deployment, possibly introducing a bias when it comes to quantifying the overall roughness of a glacier. The recent development of airborne techniques, such as uncrewed aerial vehicle (UAV) photogrammetry and airborne LiDAR, opened up new possibilities for mapping surface roughness properties. While these techniques enable the high resolution mapping of roughness obstacles, they often only cover portions of a glacier or ice sheet. On the other hand, satellite altimetry provides the means to cover entire ice sheets, though the horizontal resolution remains a limiting factor when

mapping all the obstacles that contribute to form drag. Depending on the type of surface, parameterizations using available satellite products are possible, as presented for Arctic sea-ice by Lüpkes et al. (2013), Petty et al. (2017), and Nolin and Mar (2019).

The third and final challenge is the experimental validation of bulk drag models over remote rough ice areas, which either requires in situ eddy-covariance or multi-level wind and temperature measurements. Long-term and continuous datasets re-

main scarce on the GrIS, although simplifying in situ methods can be applied for long-term monitoring of turbulent fluxes (Van Tiggelen et al., 2020, T20).

In this paper, we address the first two challenges by applying the model of Raupach (1992) to 1 m resolution elevation profiles measured over the west GrIS by the ICESat-2 laser altimeter. We apply the bulk drag model to roughness information from UAV photogrammetry, and address the third challenge by evaluating the modelled aerodynamic roughness against in situ

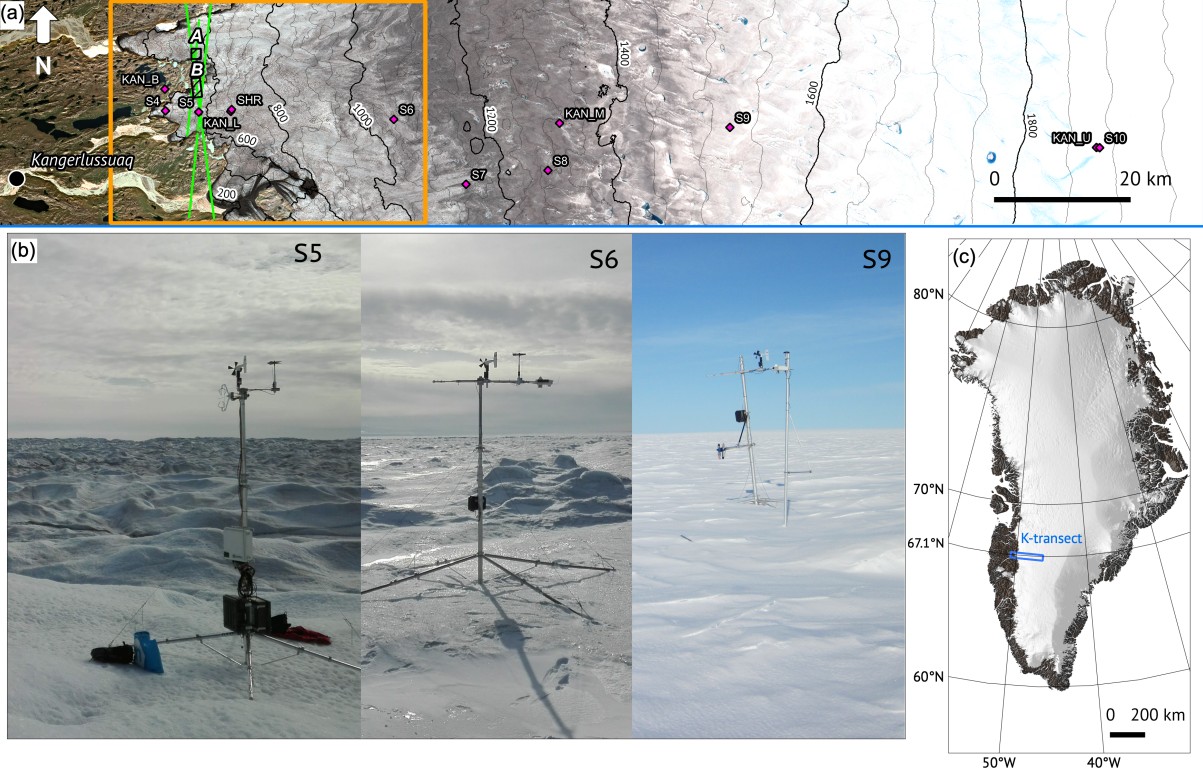

**Figure 1.** (a) Map of the K-transect, with the location of the automatic weather stations and mass balance sites indicated by the pink diamonds. The black boxes A and B delineate the areas mapped by UAV photogrammetry. The large black box indicates the area covered in Figs. 6 and 9. The background image was taken by the MSI instrument (ESA, Sentinel-2) on 12-08-2019. Pixel intensity is manually adjusted over the ice sheet for increased contrast. The green solid lines denote the ICESat-2 laser tracks that are compared to the UAV surveys (Table 2). (b) Sites S5 (06 Sep 2019), S6 (06 Sep 2019) and S9 (03 Sep 2019) taken during the yearly maintenance. Note that no data from the the AWS shown at S9 is used in this study. (c) Location of the K-transect on the Greenland Ice Sheet.

eddy-covariance measurements. We then evaluate the ICESat-2 elevation profiles against UAV photogrammetry, and finally apply the bulk drag model to the ICESat-2 profiles obtained over an extended area and during different time periods.

This paper is organised as follows. In Sect. 2 we describe the modifications in the bulk drag model, and in Sect. 3 we describe the elevation datasets used to force the model. We then evaluate the bulk drag model for one site in Sect. 4.1, and the roughness statistics derived from ICESat-2 at multiple sites in Sect. 4.3. In Sect. 4.4 we apply the model to map the aerodynamic roughness length ($z_{0m}$) along the K-transect, on the west GrIS.

## 2   Model

### 2.1   Definition of the aerodynamic roughness length $z_{0m}$

Atmospheric models assume that the lowest grid point above the surface is located in the inertial sublayer (or surface layer). In this layer, the eddy diffusivity for momentum increases linearly with height and decreases with atmospheric stability, which yields the semi-logarithmic vertical profile of horizontal wind speed. Over a rough surface, the pressure drag force on the obstacles acts as an additional sink of momentum, next to skin friction. Furthermore, the turbulent wakes generated by the flow separation enhance turbulent mixing. This may be approximated by an increase of the eddy diffusivity in the roughness sublayer (Garratt, 1992; Raupach, 1992; Harman and Finnigan, 2007). As such, the vertical profile of horizontal wind speed ($u(z)$) over a rough surface can be written as:

$$u(z) = \frac{u_*}{\kappa}\left[\ln\left(\frac{z-d}{z_{0m}}\right) - \Psi_m\left(\frac{z-d}{L_o}\right) + \Psi_m\left(\frac{z_{0m}}{L_o}\right) + \widehat{\Psi_m}(z)\right], \tag{1}$$

where $z$ is the height above the surface, $u_* = \left(\frac{\tau}{\rho}\right)^{0.5}$ is the friction velocity, $\rho$ the air density, $\kappa = 0.4$ is the von Kármán constant, $\tau$ the total surface shear stress and $z_{0m}$ is the roughness length for momentum. The average wind profile in Eq. (1) is shifted upwards by a displacement height $d$, which is defined as the centroid of the drag force profile on the roughness elements (Jackson, 1981). $z_{0m}$ is thus defined as the height above $d$ where $u(z) = 0$. The dependency of the eddy diffusivity for momentum on the diabatic stability and on the turbulent wake diffusion are described as $\Psi_m\left(\frac{z-d}{L_o}\right)$ and $\widehat{\Psi_m}(z)$, respectively, where $L_o$ is the Obukhov length. The hat notation is used for the roughness layer quantities, as in Harman and Finnigan (2007). Above the roughness sublayer, $\widehat{\Psi_m}(z) = 0$.

The problem we address is the estimation of $z_{0m}$. Rewriting Eq. (1) and assuming neutral conditions (i.e. $\Psi_m = 0$), yields :

$$z_{0m} = (z-d)\left[\exp\left(\kappa\frac{u(z)}{u_*} - \widehat{\Psi_m}(z))\right)\right]^{-1}. \tag{2}$$

Hence, the process of finding $z_{0m}$ is equivalent to finding $d$, $\frac{u(z)}{u_*}$ and $\widehat{\Psi_m}(z)$ simultaneously.

### 2.2   Bulk drag model of $z_{0m}$

The main task is to model the total surface shear stress $\tau = \rho u_*^2$, which for a rough surface is the sum of both form drag $\tau_r$ and skin friction $\tau_s$:

$$\tau = \tau_r + \tau_s. \tag{3}$$

Both $\tau_r$ and $\tau_s$ are parameters of the flow, but can be related to the geometry of the roughness obstacles using a bulk drag model. Two important parameters of the roughness obstacles are their height ($H$) and their frontal area index ($\lambda$), defined as:

$$\lambda = \frac{A_f}{A_l}, \tag{4}$$

with $A_f$ the frontal area of the roughness obstacles perpendicular to the flow, and $A_l$ the total horizontal area.

At this point, we will differ from the model by Shao and Yang (2008), who add an extra term in Eq.(3) in order to separate the skin friction at the roughness elements and the underlying surface. We also differ from the models by Lüpkes et al. (2012) and Lüpkes and Gryanik (2015), where skin friction over sea-ice is separated between a component over open water, and a component over ice floes. In the case of a rough ice surface, their is no clear distinction between the obstacles and the underlying surface. Therefore, we follow the model of Raupach (1992, R92), which is designed for surfaces with a moderate frontal area index ($\lambda < 0.2$). As we will see in the next sections, the ice surfaces considered here do not exceed $\lambda = 0.2$. As a comparison we will also consider the models from Lettau (1969, L69) and Macdonald et al. (1998, M98). The detailed equations of the bulk drag models can be found in Appendix A.

## 2.3 Definition of the height ($H$) and frontal area index ($\lambda$) over a rough ice surface

Here we introduce the type of surfaces that we are considering. Our aim is to model the aerodynamic roughness of a rough ice surface, including its dependence on wind direction (Van Tiggelen et al., 2020). We will consider rectangular elevation profiles of length $L = 200$ m, measured upwind from a point of interest (e.g. an automatic weather station, or AWS). This geometry is a strong simplification of the true fetch footprint, which is calculated for a specific wind direction at S5 in Fig. 2, after Kljun et al. (2015). Yet this simplification allows us to use 1D elevation datasets, such as profiles from the ICESat-2 satellite laser altimeter. Besides, the true fetch footprint depends on flow parameters such as the friction velocity ($u_*$) and the boundary-layer height (Kljun et al., 2015), which are not known a priori.

Four measured elevation profiles, and a high-resolution orthomosaic image are shown in Fig. 2. These were measured on 6 September 2019 at site S5 ($67.094°$ N, $50.069°$ W, 560 m) in the locally prevailing wind directions, using UAV photogrammetry, of which the details will be given in Sect. 3. At this site, pyramidal ice hummocks with heights between 0.5 m to 1.5 m are superimposed on larger domes of more than 50 m in diameter (see also Fig. 1b). The elevation profiles for different fetch directions illustrate three important issues: (1) the zero-referencing of the surface , (2) the identification of distinct roughness obstacles and (3) the important variability of the surrounding topography, depending on the fetch direction. The obstacles being anisotropic, the surface appears rougher in the southerly directions than in the easterly directions. Besides, the ice ridges and troughs have variable heights and depths, which means that describing this rough ice surface with a few length scales (e.g. $H$, $\lambda$) in order to estimate the aerodynamic roughness will introduce some uncertainty. This is mainly because each individual obstacle has a different contribution to the total drag. Unfortunately, these individual drag contributions cannot be modelled, due to the unknown shape of the wind profile between the roughness elements. Without a universal theory of drag over complex surfaces, several simplifications need to be made.

We choose here to approximate the true surface as an array of $f$ identical obstacles of height $H$ in the profile of length $L$ (Munro, 1989; Smith et al., 2016) (Fig. 3). This avoids the use of empirical formulas for the estimation of $z_{0m}$, and allows us to apply the bulk drag models. The approach of approximating a natural surface by uniquely shaped obstacles is formally justified by Kean and Smith (2006), as most of the form drag is caused by the largest and steepest obstacles. On the other hand, large natural obstacles also tend to be wide, so their relatively small frontal area index considerably reduces their contribution to the

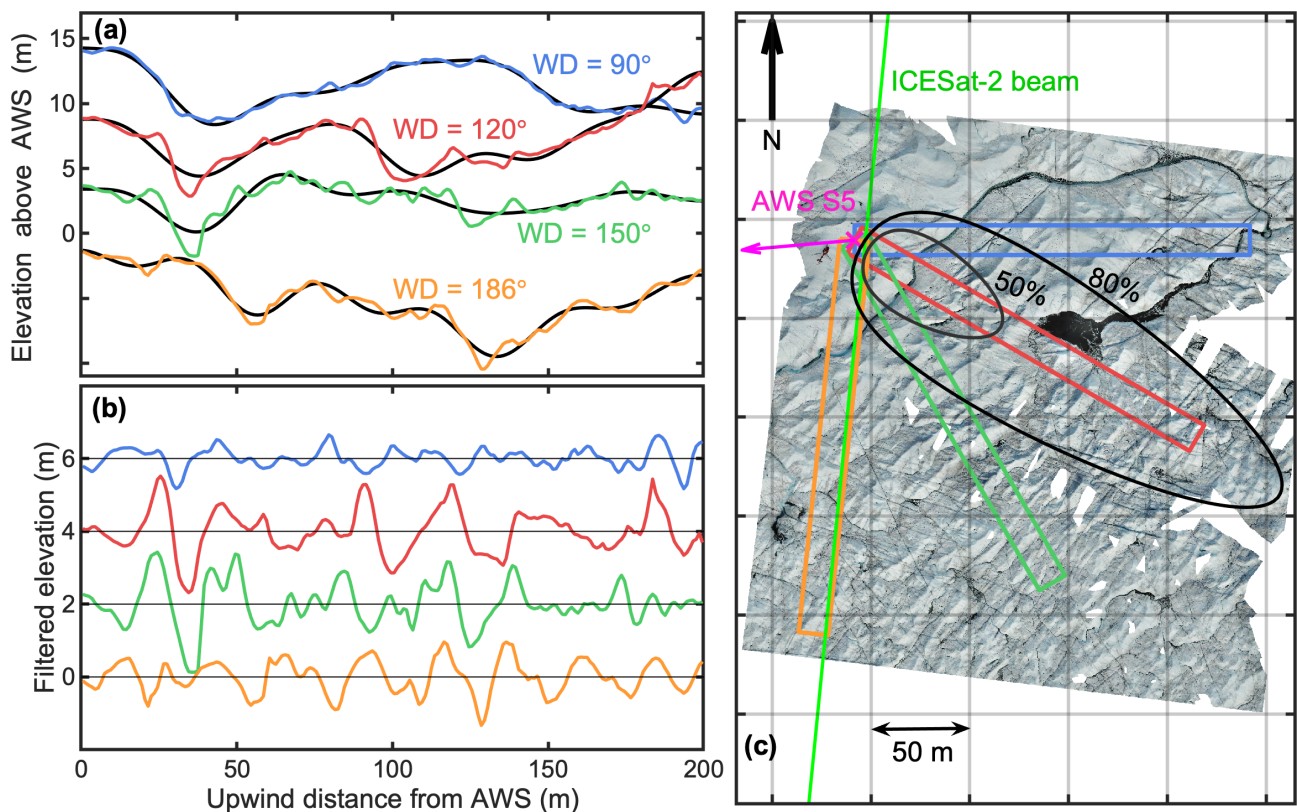

**Figure 2.** (a) Measured elevation profiles for four different wind directions upwind of AWS S5, (b) Filtered elevation profiles and (c) orthomosaic true-color image of AWS S5 and surroundings taken by UAV photogrammetry on 6 September 2019. The different coloured rectangles in (c) indicate the profiles shown in panel (a). The profiles have been vertically offset by 5 m in (a) and by 2 m in (b) for clarity. The black line in (a) denotes the low-frequency contribution of the profiles for a cut-off wavelength $\Lambda = 35$ m. The pink arrow in (c) denotes the displacement vector of the AWS between the ICESat-2 overpass on 14 March 2019 and the UAV imagery on 6 September 2019. The estimated extent of the 50% and 80% fetch footprints for the data in Sep 2019 in wind directions $\in [179; 181]^{\circ}$ is shown by the black ovals.

total form drag (Fitzpatrick et al., 2018). To remove the influence of the widest obstacles, the elevation profile of length $L$ is linearly detrended and the power spectral density of the detrended profile is computed in order to filter out all the wavelengths larger than the cutoff wavelength $\Lambda = 35$ m. This value is found to give optimal results, which is shown in Appendix B. In order to avoid spectral leakage when applying Fourier statistics on short and aperiodic signals, we extend each input profile with the identical but mirrored profile before computing the power spectral density. This yields a symmetrical thus periodic profile of length $2L$, which is then high-pass filtered. The final statistics are then computed using the first half of the filtered profile of length $L$. Typical filtered profiles are shown in Fig. 2b and Fig. 3. These profiles only contain the ice hummocks, as the high-pass filter removed the influence of the large-scale domes.

125

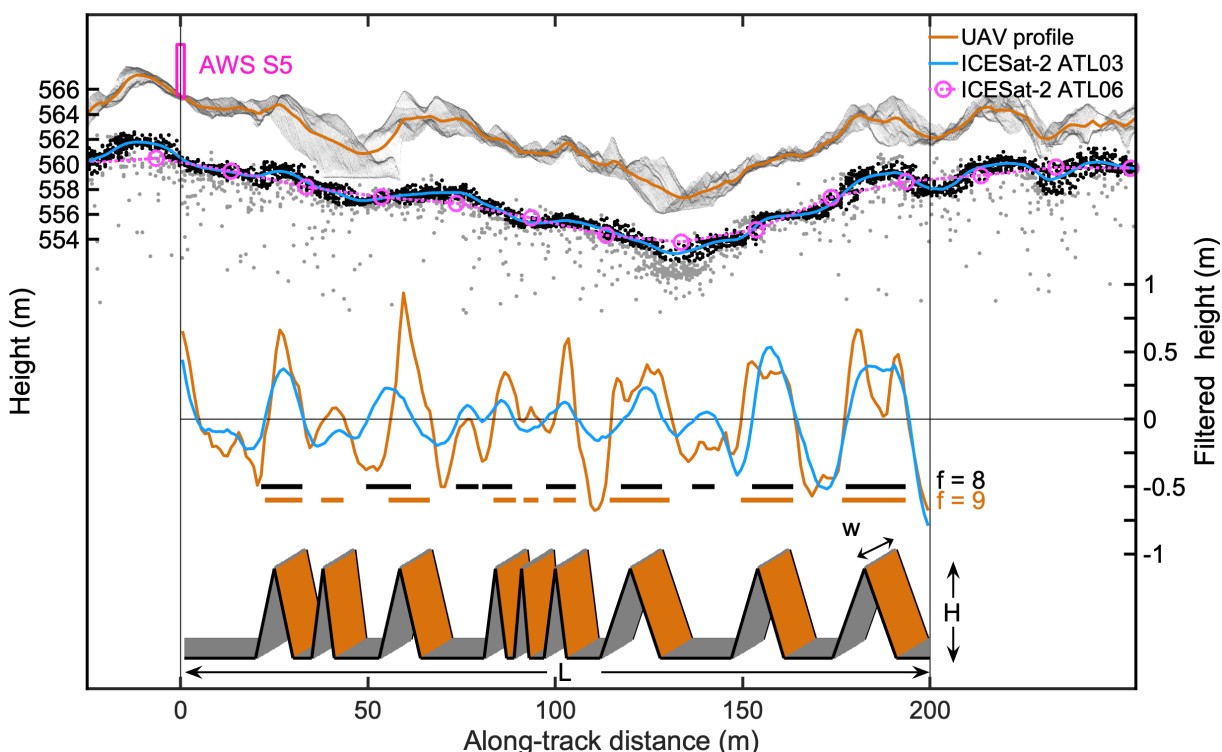

**Figure 3.** Steps in converting a measured digital elevation model to the modelled topography, where $L$ is the length of the profile, $f$ the number of obstacles, $H$ the height of the obstacles and $w$ the width of the elevation profile. The location and height of AWS S5 is shown on top of the UAV elevation profile. The black dots denotes all the ATL03 photons, while the grey dots denote the selected photons for the kriging procedure. The solid black line denotes the 1 m resolution interpolated profile for ATL03 data, and the pink dots denote the 20 m resolution ATL06 signal.

The height of the roughness obstacles ($H$) is taken as :

$$H = 2\widetilde{\sigma_z}, \tag{5}$$

in which $\widetilde{\sigma_z}$ is the standard deviation of the filtered elevation profile. This is an arbitrary but convenient choice, as the standard deviation of the topography captures all the scales in the filtered profile but remains insensitive to the height of the small-scale obstacles, which we assume to have a negligible influence on the overall drag. Unfortunately, the variance is sensitive to the height of the largest obstacles, and thus to the chosen value for $\Lambda$.

Next, we define an obstacle as a group of consecutive positive values of filtered heights, after Munro (1989) (see also Smith et al. (2016)), which yields $f$, the number of obstacles. The obstacle frontal area index ($\lambda$) in the direction of the elevation profile is then computed as (Fig. 3),

$$\lambda = f\frac{Hw}{Lw} = \frac{fH}{L}, \tag{6}$$

where $w$ is the width of the profile, set to 15 m. This value was chosen to match the approximate ICESat-2 footprint diameter, yet it is much smaller than width of the real fetch footprint (Fig. 2). We assume that the obstacles and the elevation profile have the same width, which removes all information about the shape of the obstacles in the direction perpendicular to the wind direction. This simplification avoids the additional uncertainty regarding the aggregation of 2D datasets in the process of modelling $z_{0m}$, and allows us to apply the model to ICESat-2 profiles.

To summarize, a measured elevation profile is now completely defined by the height of the obstacles ($H$), and the frontal area index ($\lambda$), after high-pass filtering (see Fig. 3). This now allows us to apply a bulk drag model to estimate one value for $z_{0m}$ per 200 m profile. The exact placement of the obstacles is resolved in the process (Fig. 3), but does not serve as input for the drag models. Detailed equations of the bulk drag model can be found in Appendix A.

## 3 Datasets

### 3.1 Eddy-covariance measurements

Vertical propeller eddy covariance (VPEC, see also T20) measurements are available at sites S5 ($67.094°$ N, $50.069°$ W, 560 m) and S6 ($67.079°$ N, $49.407°$ W, 1010 m) since 2016, while AWS observations are available since 1993 and 1995 for each site Smeets et al. (2018). For this study we use eddy-covariance measurements acquired during September 2019 at site S5 and also site SHR ($67.097°$ N, $49.957°$ W, 710 m), and during from September 2018 to August 2019 at site S6. All these sites are situated in the lower ablation area of the K-transect, which is a 140 km transect of AWS and mass balance observations on the western part of the GrIS (Van de Wal et al., 2012; Smeets et al., 2018). It extends from the ice edge up to 1850 m elevation, and therefore covers many contrasting types of surfaces, ranging from the rough crevassed bare ice close to the ice edge, to the year-round firn-covered surface at the highest locations (see Figs. 1 and 6). At the end of the melting season, the bare ice surface at S5 and SHR is characterised densely packed hummocks up to 1.5 m height, while at S6 it is characterised by more sparsely packed hummocks of 0.6 m average height. The datasets include 30-min observations of the friction velocity $u_*(z)$, and wind speed $u(z)$ at the same height above the surface ($z = 3.7$ m). Two independent techniques were used at S5 and SHR. The first technique is the sonic eddy-covariance (SEC) method, which uses measurements from a sonic anemometer (CSAT3B, Campbell scientific, Logan, USA) sampled at 10 Hz. The second technique is VPEC method, that relies on measurements of a vertical propeller, horizontal propeller and fine-wire thermocouple sampled at 5 Hz. At S6, only the VPEC method was used with a sampling interval of 4 s. For both methods, the roughness length ($z_{0m}$) is calculated using Eq. (2).

We only select data taken during near-neutral conditions ($z/L_o < 0.1$), and we assume that the measurements are taken above the roughness layer, i.e. $\widehat{\Psi_m(z)} = 0$. The latter is a reasonable assumption, given that the height of the obstacles ($H$) at these sites is less than 1.5 m, which means that the roughness layer unlikely exceeds 3 m (Smeets et al., 1999; Harman and Finnigan, 2007). On the other hand, when applying the drag model to estimate $z_{0m}$ (Appendix A.), the correction factor $\widehat{\Psi_m(z)}$ is taken into account. The reason is that the obstacles are located in the roughness layer, where the vertical wind profiles deviate from the inertial sublayer wind profiles, according to Eq. (1). Details about the processing steps and further data selection strategies can be found in T20. The data selection strategy removes all data points with wind directions outside the $[80°; 200°]$ interval. In

**Table 1.** Description of $z_{0m}$ in situ datasets on the K-transect. The data from Meesters et al. (1997), Smeets and Van den Broeke (2008), Lenaerts et al. (2014) and Van Tiggelen et al. (2020) are denoted M97, SB08, L14 and T20 , respectively. The measurement methods are (p) profile, (sec) sonic eddy-covariance or (vpec) vertical propeller eddy covariance.

| site (elevation) | season | surface type | $z_{0m}$ range (m) | reference (method) | data averaging period |
|---|---|---|---|---|---|
| S5 (550 m) | summer | densely-packed ice hummocks, between 0.5 m and 1.5 m | $6 \times 10^{-3} - 8 \times 10^{-2}$ | SB08 (p) | Sep 2003 & Aug 2004 |
| | | | $1.79 \times 10^{-3} - 3.45 \times 10^{-2}$ | T20 (vpec) | Sep 2016 |
| | | | $1.05 \times 10^{-2} - 4.66 \times 10^{-2}$ | this study (sec) | Sep 2019 |
| | winter | snow and exposed ice hummocks | $8 \times 10^{-5} - 1 \times 10^{-3}$ | SB08 (p) | Dec 2003 to May 2004 |
| | | | $2.9 \times 10^{-4} - 1.21 \times 10^{-2}$ | T20 (vpec) | Dec 2016 to May 2017 |
| SHR (710 m) | summer | densely-packed ice hummocks, between 0.5 m an 1 m | $9.1 \times 10^{-3} - 4.38 \times 10^{-2}$ | this study (sec) | Sep 2019 |
| S6 (1010 m) | summer | sparse ice hummocks, average height 0.6 m | $2 \times 10^{-3} - 2 \times 10^{-2}$ | SB08 (sec) | Sep 2003 & Aug 2004 |
| | | | $1.26 \times 10^{-3} - 7.52 \times 10^{-3}$ | this study (vpec) | Aug 2019 |
| | winter | snow, sastrugi | $5 \times 10^{-5} - 6 \times 10^{-4}$ | SB08 (sec) | Dec 2003 to May 2004 |
| | | | $2 \times 10^{-6} - 1.33 \times 10^{-4}$ | this study (vpec) | Dec 2018 to May 2019 |
| S9 (1520 m) | summer | changing from wet melting snow to large ice crystals | $2 \times 10^{-6} - 1 \times 10^{-4}$ | SB08 (p) | Sep 2003 & Aug 2004 |
| | | | $2 \times 10^{-4} - 5 \times 10^{-4}$ | M97 (sec) | July 1991 |
| | winter | snow, sastrugi | $2 \times 10^{-5} - 7 \times 10^{-4}$ | SB08 (p) | Dec 2003 to May 2004 |
| S10 (1880 m) | summer | snow, sastrugi | $2 \times 10^{-4} - 7 \times 10^{-4}$ | L16 (sec) | Sep 2012 |

the following sections we average $\ln(z_{0m})$, which is of interest for the determination of the vertical profile of horizontal wind speed (Eq. (1)). Additional in situ averaged $z_{0m}$ measurements obtained during different time periods and at several locations along the K-transect are taken from Meesters et al. (1997, M97), Smeets and Van den Broeke (2008, SB08), Lenaerts et al. (2014, L14) and T20 and are summarized in Table 1.

## 3.2 UAV structure from motion

The high-resolution elevation maps are derived using a structure-from-motion workflow using UAV imagery. Two crevassed areas close to the ice edge were mapped using an eBee fixed-wing UAV from Sensefly, while the area surrounding S5 was mapped using a Mavic Pro quadcopter UAV from DJI. Multiple overlapping true-color images of the surface are processed in Agisoft Photoscan to produce 3D elevation maps. Detailed information about this workflow can be found in Immerzeel et al. (2014), Kraaijenbrink et al. (2016) and references therein. Briefly, the same surface features are identified on different images and are used to reconstruct the 3D geometry between the surface and the camera position. The resulting point cloud of the surface is then gridded and finally geo-referenced using the information of the UAV GPS, which yields a digital elevation

**Table 2.** Description of DEMs obtained by UAV photogrammetry, and of the corresponding overlapping ICESat-2 laser beams.

| Site | Center coordinate | Dimensions (m) | Resolution (m) | UAV survey date | ICESat-2 track-cycle-beam | ICESat-2 date |
|------|-------------------|----------------|----------------|-----------------|---------------------------|---------------|
| A | 67.171 N, 50.075 W | 1500 x 1400 | 0.3 | 01-09-2019 | 1169-04 gt1r | 12-09-2019 |
| B | 67.126 N, 50.075 W | 2300 x 1300 | 0.3 | 03-09-2019 | 1344-05-gt1r | 23-12-2019 |
| S5 | 67.093 N, 50.065 W | 450 x 375 | 0.025 | 06-09-2019 | 1169-02-gt1l | 14-03-2019 |

model (DEM) of the surface. No additional ground-control points were used for the elevation maps used in this study, which is of little relevance in this study, as we are not interested in the exact absolute elevation but in relative obstacle heights. Details about the UAV DEMs are provided in Table 2.

The elevation profiles are then extracted by projecting all the DEM points in a 200 m x 15 m rectangle on the center line, followed by averaging the projected points in 1 m bins (see Fig. 3). The aim of this averaging method is to mimic ICESat-2 profiles.

## 3.3 ICESat-2 laser altimeter

Launched in September 2018 by the National Aeronautics and Space Administration (NASA), the ICESat-2 (Ice, Cloud, and Elevation Satellite-2) satellite carries a laser altimeter system in near-polar orbit (Markus et al., 2017). The altimeter relies on a photon-counting system, which in combination with both the spacecraft's position and its pointing orientation, enables the retrieval of 3-D position of individual backscattered photons (Neumann et al., 2019). Our hypothesis is that the small footprint diameter ($\approx$ 15 m) and short along-track spacing between these footprints (0.7 m) allows for an accurate estimation of land ice aerodynamic roughness properties.

A typical geolocated photon measurement ATL03 (Neumann et al., 2019) can be seen in Fig. 3 for site S5, and in Fig. 4a for area A. Details about which ICESat-2 measurements are compared against the UAV surveys are provided in Table 2. Not more than one ICESat-2 measurement exactly overlaps each UAV survey. This is mainly due to the presence of clouds and due to changes in laser pointing orientations in other ICESat-2 measurements, but also due to changes in the studied locations due to ice flow. The global geolocated photon product ATL03 requires some processing steps before the roughness statistics can be computed. These steps mainly involve the selection of valid photons, aggregating the 3-D photon positions on a regular along-track grid, and finally correcting for remaining biases. The standard algorithm used to derive an accurate estimate of land ice height product from ATL03 to ATL06 is described in detail by Smith et al. (2019). Unfortunately the 20 m along-track resolution of the ATL06 land ice height product is too coarse for aerodynamic roughness calculations for two reasons. First, in the ATL06 product there are only ten points in 200 m sections, which is not enough to apply the high-pass filter. And second, on this scale the amount of roughness obstacles ($f$) would be greatly underestimated as can be seen in Figs. 3 & 4. Fortunately, information smaller than the footprint diameter can be extracted from the ATL03 product, as shown by Herzfeld et al. (2020), in which a density-dimension algorithm is used that facilitates surface-height determination at the 0.7 m nominal along-tack

resolution. In the following part we describe a method to produce a 1 m resolution along-track surface height estimation from the ATL03 raw photons signal.

The first step involves selecting all the ATL03 photons that have been flagged as either low, medium or high confidence by the ATL03 algorithm. All the selected photons are projected on the along-track segment, and a median absolute difference filter
is used to remove all the photon heights which deviate too much from the local ensemble median,

$$\langle z \rangle - \frac{q_{low}}{0.6745} \langle |z - \langle z \rangle | \rangle \leq z \leq \langle z \rangle + \frac{q_{high}}{0.6745} \langle |z - \langle z \rangle | \rangle, \tag{7}$$

where $< z >$ denotes the median of $z$ within a moving window. We choose $q_{low} = 1$ and $q_{high} = 2$, in order to filter more photons below than above the median. We assume that the highest detected photons are more likely to be first surface reflections, while the lower photons are more likely to be delayed by scattering. We set the window length to 50 m. The previous selection
strategy could also be applied for retrieving the surface in the case of multiple reflections (e.g. shallow supraglacial lakes), but this was not tested.

The second step involves interpolating the irregular photon locations on a regular, 1 m resolution, along-track grid. The overlap between the individual footprints means that the geolocated photon heights in close vicinity must be correlated to each other, with a correlation diameter similar to twice the footprint diameter ($\approx$ 30 m). We take advantage of this feature to
interpolate the ATL03 photons using a k-nearest neighbour, one dimensional, ordinary kriging algorithm, of which details can be found in e.g. Hengl (2009). In essence, the interpolation weights depend on the covariance with the nearby measurements, which is assumed to decrease over distance. A gaussian covariance function with a radius of 15 m is found to fit the experimental semi-variograms best. For computational efficiency, only the 100 closest geolocated photons within a quarter footprint diameter (3.75 m) of each grid point are used for the interpolation. We only choose the high confidence photons, but if there is less than
1 photon per 0.7 m, we also select the medium confidence photons. If there are not enough medium confidence photons, we increase the search radius to half the footprint diameter (7.5 m) or even up to a footprint diameter (15 m). The low confidence photons are only used as a last resort. If in a 15 m footprint diameter there are still not enough photons present, the height on that grid point is not estimated, which results in a gap. A sensitivity experiment using different photon selection strategies and different kriging parameters is found in Appendix B.
The last step involves grouping the interpolated elevation measurements in 200 m along-track windows, and the high-pass filtering using a cut-off wavelength of $\Lambda = 35$ m (Sect. 2). The height of the obstacles ($H$) is defined as twice the standard deviation of the filtered signal (Eq. (5)).

Although 1 m resolution is still too coarse to capture all the small-scale obstacles that contribute to form drag, we expect that most of the form drag over rough ice is caused by the larger obstacles that are resolved by the ICESat-2 altimeter. Furthermore,
the small-scale information is still indirectly present in the scatter of the surrounding photons to the closest grid point, which is a measure of both the instrumental error and the surface slope, but also of the surface roughness (Gardner, 1982).

An alternative approach that does not require gridding the ATL03 product to 1 m resolution would be to use the standard deviation of the raw photon signal detrended for the resolved 20-m resolution ATL06 data, as in Yi et al. (2005) and Kurtz

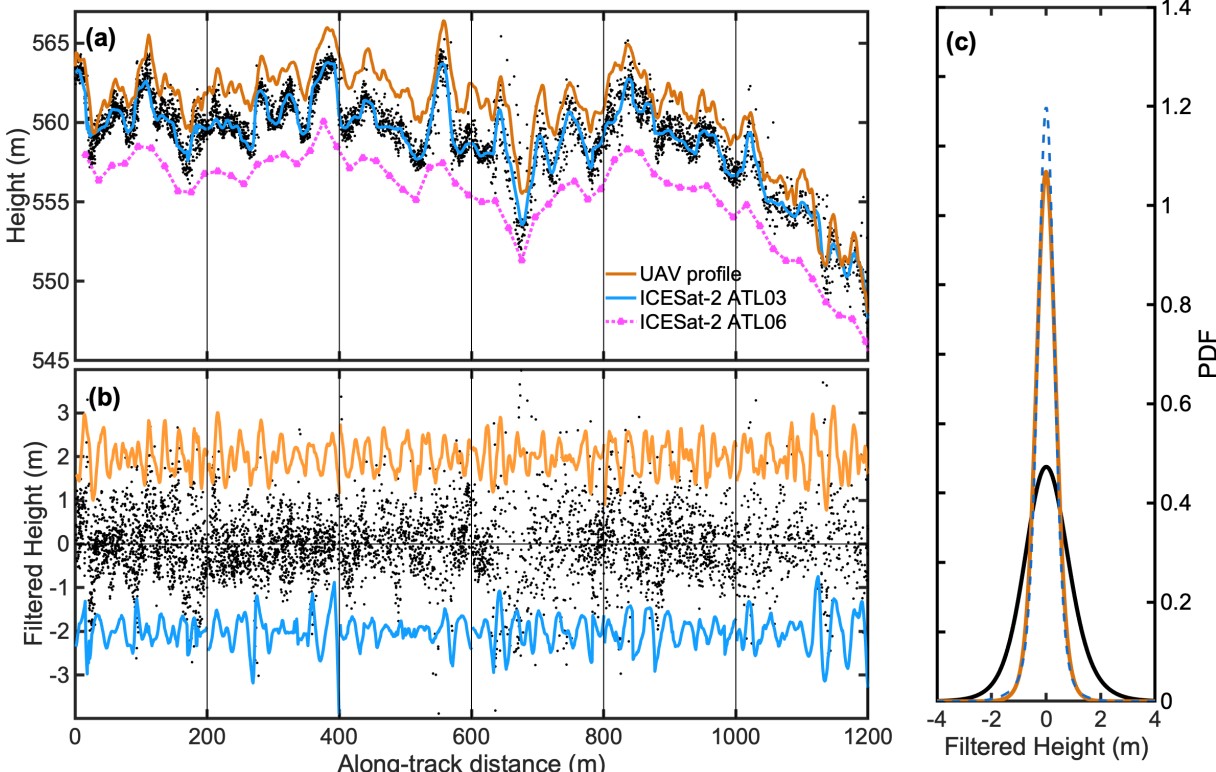

**Figure 4.** (a) Elevation profile at site A measured by the UAV and by ICESat-2 (solid lines), selected ICESat-2 photons (grey dots) and ICESat-2 ATL06 height (pink dashed line). The UAV and ATL06 profiles have been vertically offset by 2 m for clarity. (b) Filtered profiles (solid lines) and residual photons elevations after filtering per 200 m windows (grey dots), where the UAV and ATL03 filtered profiles have also been vertically offset. (c) Probability density function of the filtered ICEsat-2 profile (black dashed line), UAV profile (orange solid line) and residual photons elevations (grey line).

et al. (2008). However, as we will see in the following sections, this would overestimate the height of the roughness obstacles.
Besides, the frontal area index ($\lambda$) would remain unknown.

When working with the 1-m interpolation profile, we model the standard deviation of the unresolved topography ($\sigma_{sub}$) according to,

$$\sigma_{sub} = \left(\sigma_{ph,res}^2 - \sigma_i^2\right)^{0.5}/2, \tag{8}$$

where $\sigma_{ph,res}$ is the standard deviation of the photon residual elevations, defined as the signal of the selected photons minus the
interpolated 1 m resolution profile (Fig. 4), $\sigma_i = 0.13$ m is the standard deviation due to the instrumental precision (Brunt et al., 2019). We calculate $\sigma_{sub}$ for each 200 m profile. The total variance of the surface elevation measured by the laser altimeter in 200 m intervals is the sum of both the resolved and unresolved variance:

$$\sigma_{tot} = \sqrt{\widetilde{\sigma_{res}}^2 + \sigma_{sub}^2} \tag{9}$$

in which $\widetilde{\sigma_{res}}$ is the resolved standard deviation of the filtered 1 m resolution profile. The height of the roughness obstacles, corrected for the unresolved topography is then estimated according to:

$$H_{corr} = 2\sigma_{tot} \tag{10}$$

The obstacle frontal area index ($\lambda$) is finally computed using Eq. (6), where the number of obstacles ($f$) is estimated from the filtered profiles. Both $H$ and $\lambda$ are then used as input for the bulk drag model (Appendix A), which results in one value for $z_{0m}$ per 200 m profile.

The filtered ICESat-2 signal and residual photon elevations at site A is shown in Fig. 4b, and their probability density functions in Fig. 4c. At this site, the filtered ICESat-2 signal at 1 m resolution captures most of the information present in the UAV signal. On the other hand, the residual photon elevations, defined as the selected photons detrended for the interpolated profile under Eq. (9) still contain much larger scatter than the UAV elevation profile. This demonstrates that roughness is not the only factor explaining the scatter in the raw altimeter signal. Therefore using the residual scatter (Eq. (9)) will overestimate the height of the roughness obstacles. In the next sections, we will analyse the uncorrected height of the obstacles ($H$), unless stated otherwise.

## 4   Results

### 4.1   Evaluation of the bulk drag model forced with a UAV DEM

Bulk drag models are often used as a convenient way to estimate the aerodynamic properties of natural surfaces. Nevertheless, the number of quantitative evaluations of these models for rough snow and ice surfaces is very limited. Brock et al. (2006) found that $z_{0m}$ modelled using the method by L69 (Eq.(A13)) agrees well with observations over melting ice on a mountain glacier, although they used shorter profiles, up to 15 m in length, and sampled in the orientation perpendicular to the wind direction. On the other hand, Van den Broeke (1996) found that L69 overestimates $z_{0m}$ at site S4 at the K-transect (lowest site in Fig. 6). The same overestimation was found by Smeets et al. (1999), Fitzpatrick et al. (2018) and Chambers et al. (2019) for rough glacier ice, but also by Miles et al. (2017) for a debris-covered glacier. These studies all use different methods at different sites to estimate $H$ and $\lambda$, which illustrates the limited suitability of the model by L69 for realistic snow and ice surfaces.

To verify the suitability of several drag models (see Appendix A), we use the eddy-covariance observations at site S5 as independent validation (Sect. 3). Different values of $z_{0m}$ are calculated for different fetch directions as depicted in Fig. 2. Figure 5 compares both the estimated $z_{0m}$ from in situ observations and the modelled $z_{0m}$ at the end of the ablation season, as a function of the measured obstacle frontal area index $\lambda$. The L69 model (Eq.(A13)) overestimates $z_{0m}$ for $\lambda < 0.04$ at this location (Fig. 5, blue line). In accord with L69, the drag coefficient of an individual obstacle $C_d = 0.25$ is likely too high for naturally streamlined obstacles. Furthermore, L69 does not consider the displacement height, which means that the height of the obstacles ($H$) relevant for form drag is overestimated. Nevertheless, L69 still yields a reasonable estimate of $z_{0m}$ for $\lambda > 0.04$, which can be explained by the neglect of the displacement that is compensated by too small $C_d$ for these fetch directions. The method by M98 (Eq. (A14)) does account for the displacement height and, while using the same drag coefficient $C_d = 0.25$,

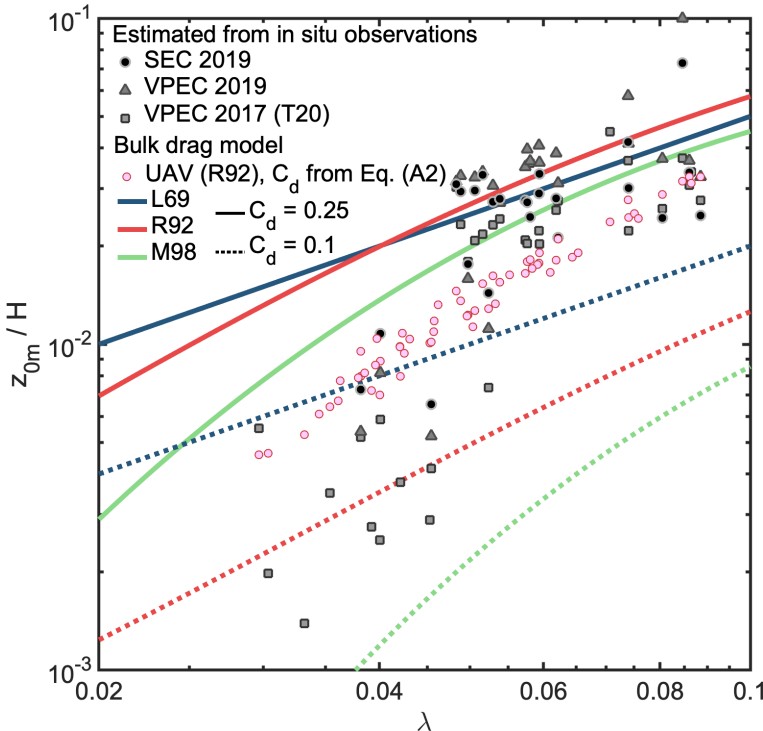

**Figure 5.** Modelled $z_{0m}$ at site S5 using three different bulk drag models: Lettau (1969, L69, blue lines), Macdonald et al. (1998, M98, green lines), Raupach (1992, R92, red lines) and using two different values for the drag coefficient for form drag: $C_d = 0.25$ (solid lines) and $C_d = 0.1$ (dashed lines). Solid grey symbols are measurements from sonic eddy-covariance (SEC) or vertical propeller eddy-covariance (VPEC). Additional data are from (Van Tiggelen et al., 2020, T20). Pink circles are the model results forced with $H$ and $\lambda$ from UAV photogrammetry, using the R92 model and $C_d$ parameterized using Eq. (A2).

it gives improved results for $\lambda < 0.04$ compared to L69 (Fig. 5, green line). The same holds for the model by R92 (Fig. 5, red line). M98 is expected to fail for very small $\lambda$, due to the absence of skin friction. Using $C_d = 0.1$, all three models perform better for $\lambda < 0.05$ but perform poorly for $\lambda > 0.04$ (Fig. 5, dashed lines). This is a strong indication that $C_d$ is not constant, but varies with the wind direction, depending on the exact placement and shape of the obstacles. In Sect. 4.3 we estimate the values
for $C_d$ required to fit the model to the observations; these values vary between 0.1 and 0.3, and show a weak relationship with $H$. The parametrization for $C_d$ from Garbrecht et al. (1999) (Eq. (A2)), for which $C_d$ increases with $H$, yields most acceptable results when used in combination with the R92 model (Fig. 5). Note that Lüpkes et al. (2012) use a constant value for $C_d$.

The R92 model with the parametrization for $C_d$ allows for some variability in modelled $z_{0m}$ for the same $\lambda$, but is still not able to reproduce the eddy-covariance observations (Fig. 5). We attribute this to the parametrization of $C_d$, that was derived
for sea-ice pressure ridges and therefore likely less suitable for rough ice hummocks. Nevertheless, the overall error between model and observation is acceptable, given the simplicity of the bulk drag models that were designed for idealized roughness geometries. As pointed out by L69, realistic modelling of total drag over a complex natural surface should intuitively require a

complete variance spectrum of the topography. Linking variance spectra to the total drag has been investigated recently through numerical simulations (Yang et al., 2016; Zhu and Anderson, 2019; Li et al., 2020), but a universal and physically based

relationship for complex surfaces is still lacking. In the next sections, we therefore use the model of R92 with a parametrized $C_d$ for mapping $z_{0m}$ using either UAV or ICESat-2 profiles.

## 4.2 Height of the roughness obstacles (H) estimated from ICESat-2

The estimated height of the obstacles ($H$) using two years of ICESat-2 measurements (16 October 2018 – 06 September 2020) crossing the lower part of the K-transect is shown in Fig. 6. $H$ ranges between less then 0.1 m at the higher locations, to more

than 3 m in rough crevassed areas near the ice edge. At first glance a clear pattern of roughness emerges, in which ice dynamics and elevation seem to be the controlling factors. Low-lying bare ice areas are rougher, while the higher, firn covered areas are smooth. Nevertheless, the roughness is very variable locally due to isolated crevasses and melt channels. Besides, we expect a seasonal variability that is not yet captured in this analysis.

## 4.3 Evaluation of ICESat-2 roughness statistics against UAV DEMs

Climate models and satellite altimeter corrections require information about the larger-scale spatial variability of surface (aero-dynamic) roughness. This motivated us to compare the roughness statistics acquired with high-resolution UAV photogrammetry to the statistics estimated from the ICESat-2 laser altimeter.

The elevation profile from the UAV survey in box A (Fig. 6) was already compared to the overlapping ICESat-2 profiles in Fig. 4a, while $H$, $\lambda$ and $z_{0m}$ are compared in Fig. 7. In box A, the UAV and ICESat-2 profiles were taken 11 days apart at

the end of the ablation season. The height ($H$) and frontal area index ($\lambda$) of the roughness obstacles is estimated for 200 m intervals, with each interval center separated by 50 m. Overall, the uncorrected 1 m profile from ICESat-2 (Fig. 7, solid black line) clearly captures all the largest obstacles and the large-scale variability, but still slightly underestimates both the height ($H$) and the frontal area index ($\lambda$) of the obstacles, compared to the UAV surveys (Fig. 7, orange line). This is expected, given the size of the laser footprint and the low-pass filtering properties of the kriging procedure. On the other hand, the correction

using the standard deviation of the photons distribution (Eq. (9)) overestimates $H$ (Fig. 7, dashed black line). This can be explained by additional processes that affect the local photon distribution but that we did not consider, such as the forward scattering in the atmosphere (Kurtz et al., 2008), the penetration of photons in the ice layer (Cooper et al., 2020), or simply the presence of outliers that passed the median absolute difference filter (Eq. (7)). Furthermore, the obstacle frontal area index ($\lambda$) is underestimated by the ICESat-2 altimeter, since we do not account for unresolved obstacles when counting the number of

obstacles ($f$). In addition, using the standard deviations of the ATL03 product detrended for the 20 m resolution ATL06 signal, results in an even greater overestimation of $H$ (Fig. 7, purple line). This is due to the fact that besides the additional processes broadening the altimeter signal, the scatter of this signal also contains the large scale variability at wavelengths larger than $\Lambda = 35$ m. We assumed that such large wavelengths can be neglected in the drag calculations, therefore they are removed in the filtered UAV and ICESat-2 profiles.

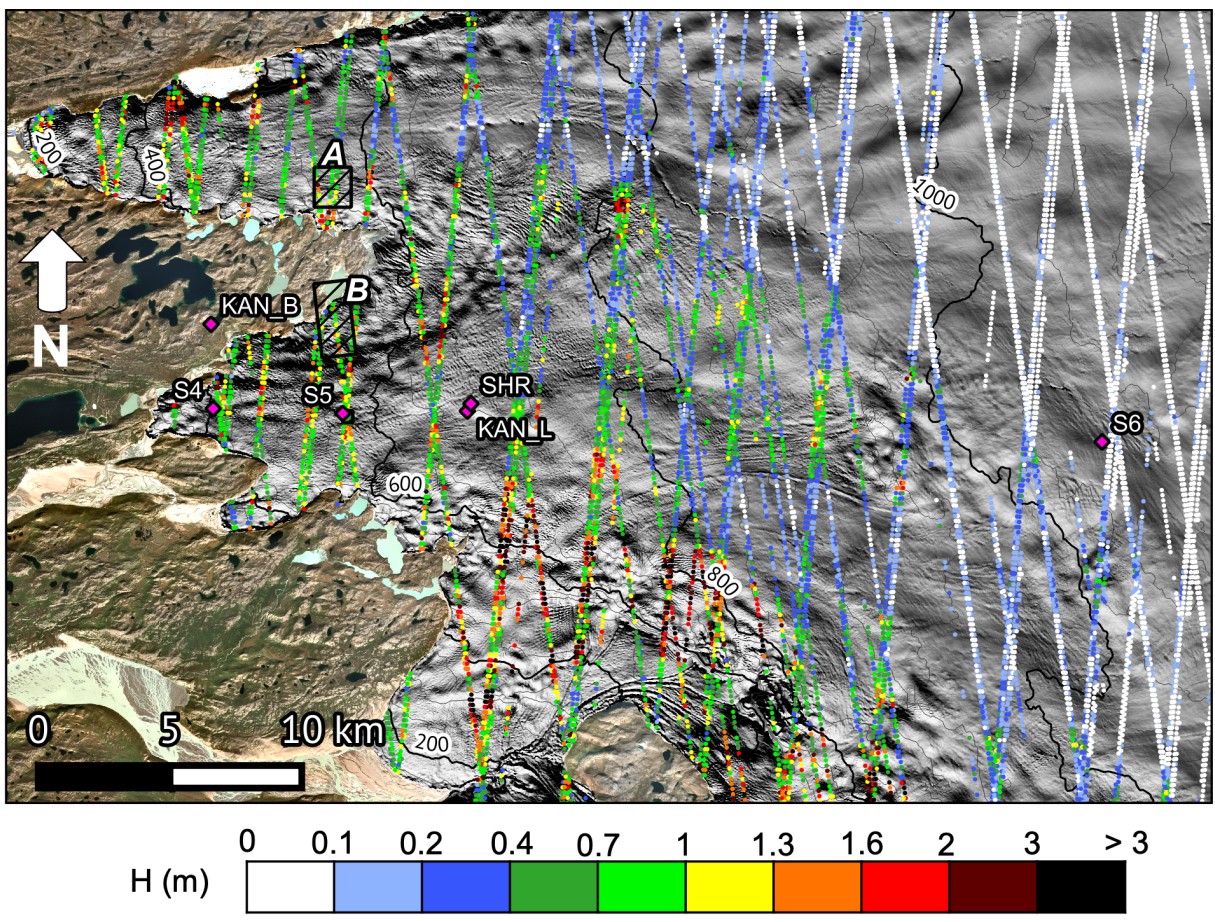

**Figure 6.** Estimated height of the roughness obstacles ($H$) from ICESat-2 between 16 October 2018 and 06 September 2020 in the lower part of the K-transect, West Greenland. The location of the automatic weather stations are given by the pink diamonds. The black boxes A and B delineate the areas mapped by UAV photogrammetry. A hillshade of ArcticDEM (Porter et al., 2018) is shown as background over the ice sheet.

Two more UAV surveys were performed in September 2019 in area B and around S5, but the overlapping ICESat-2 profiles were measured during winter (see Table 2). The comparison of $H$, $\lambda$ and modelled $z_{0m}$ is given in Fig. 7. In area B, crevassed and slightly rougher than A, the elevation was measured in December, three months after the UAV survey. The uncorrected ICESat-2 profiles show a slightly more pronounced underestimation of $H$ compared to area A, which we relate to snowfall reducing the height of the roughness obstacles. On the other hand, the corrected ICESat-2 profiles overestimate $H$ by 0.06 m, which translates in an overestimation of $z_{0m}$ by approximately $2.5 \times 10^{-3}$ m (Fig. 7). On average, the uncorrected ICESat-2 values underestimate $z_{0m}$ by $2.9 \times 10^{-3}$ m for area A and $9 \times 10^{-3}$ m for area B, which corresponds to $\approx 40\%$ and $\approx 36\%$ of the average $z_{0m}$ estimated by the UAV at these two sites, respectively.

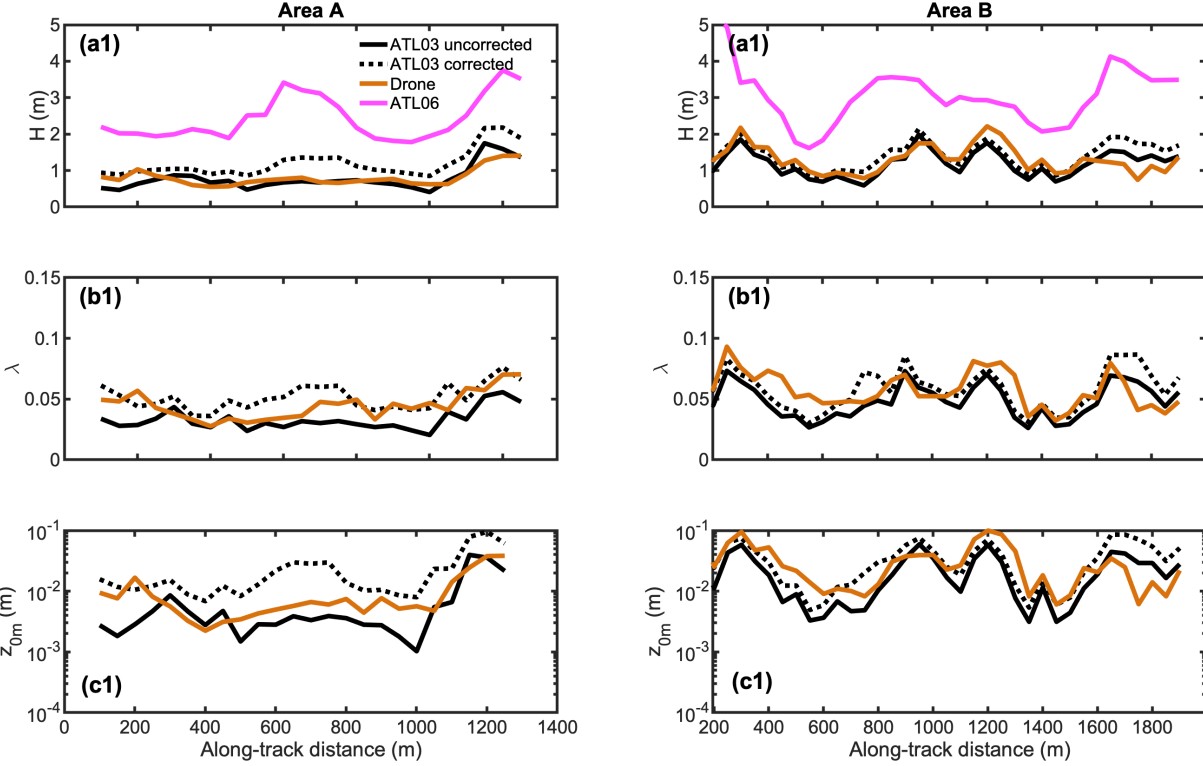

**Figure 7.** (a1) and (a2), estimated height of the roughness obstacles ($H$). (b1) and (b2), estimated frontal area index. (c1) and (c2), estimated aerodynamic roughness length ($z_{0m}$). Left panels: area A. Right panels: area B. The black lines denote the roughness statistics estimated from the ATL03 filtered profile, with or without accounting for the residual photons elevations (dashed and solid line, respectively). The orange line denotes estimates using UAV elevation profiles, and the pink line denotes the height of obstacles estimated using the scatter of ATL03 photons detrended for the ATL06 signal.

At site S5, UAV elevation profiles and eddy-covariance measurements are available in September 2019, while the ICESat-2 elevation profile was measured in March (Table 2). Both the satellite and the UAV profiles are shown in Fig. 3. Although the UAV profile is too short to statistically compare H and $\lambda$ to the ICESat-2 altimeter, the qualitative comparison between the two confirms that the satellite altimeter is very well capable of detecting most of the obstacles that are smaller than 20 m in width. Interestingly, some depressions in the UAV DEM are not captured by ICESat-2, most likely as a result of snow filling them in March. Furthermore, the bending (or 'doming') of the UAV profile is well visible near the edges, which is a consequence of the lack of reference ground control points in the UAV data processing, which is a common issue with UAV data processing (James and Robson, 2014). Both $H$ and $\lambda$ are smaller in the satellite profile than in the UAV profile, but the modelled $z_{0m}$ agrees qualitatively with that estimated using observations from the AWS S5 during March-April. During this time period, $z_{0m}$ is approximately a factor 10 smaller than during the end of the ablation season (Fig. 8, dashed orange line). Unfortunately, the track direction of the satellite altimeter rarely coincides with the wind direction measured by the anemometers at this

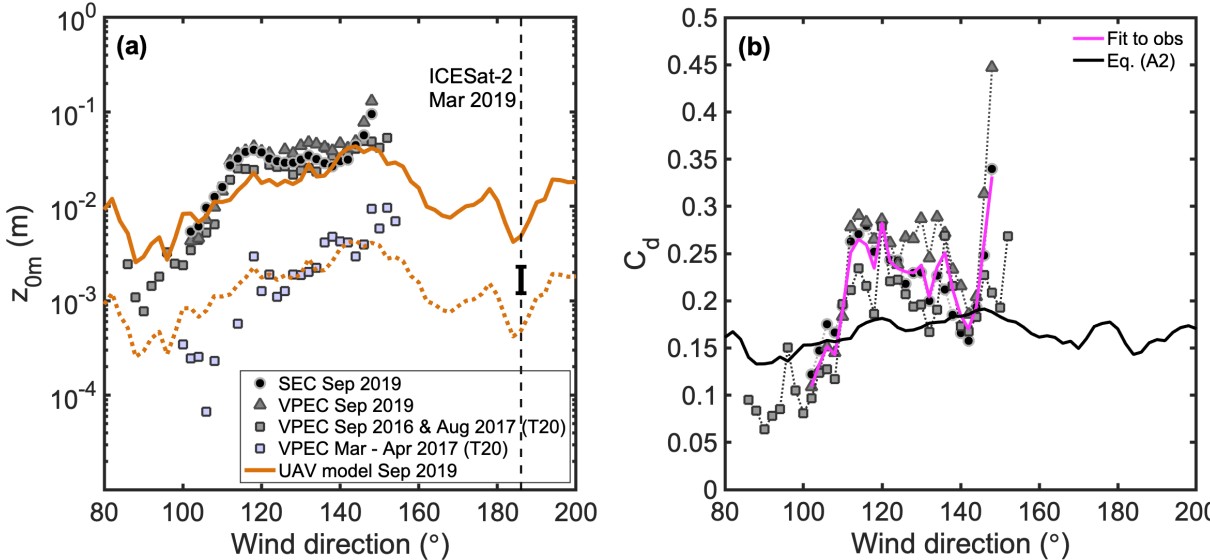

**Figure 8.** (a) Drag model evaluation at site S5. (b): Drag coefficient for form drag ($C_d$) used in the bulk drag model (black line) or required to perfectly fit the observations. The orange solid line is the modelled $z_{0m}$ using the R92 model and UAV photogrammetry on 06 September 2019, while the dashed orange line is the orange line shifted down by a factor 10. Solid symbols are measurements from sonic eddy-covariance (SEC) or vertical propeller eddy-covariance (VPEC). Additional data is from van Tiggelen et al. (2020, T20). The vertical dashed line denotes the direction sampled by the ICESat-2 laser beam on 14 March 2019. The errorbar denotes the range between the uncorrected and corrected ICESat-2 measurements.

location, due to the katabatic forcing. This prevents a direct comparison of ICESat-2 roughness to in situ observations, as
the aerodynamic roughness strongly depends on the wind direction (Van Tiggelen et al., 2020). The $z_{0m}$ value estimated from ICESat-2 profiles must thus be interpreted as the aerodynamic roughness in the wind direction along the direction of the ground laser track.

Only a high-resolution, two-dimensional DEM, e.g. obtained using a UAV, allows for an accurate description of the aerodynamic roughness around a point of interest in multiple directions. An example of such an analysis is shown for site S5 in Fig. 8.
The R92 model applied to the UAV elevation profiles reproduces the considerable variability of the estimated $z_{0m}$ using in situ observations. Across all the wind directions available in the measurements, $z_{0m}$ using the UAV profiles is underestimated by $7.6 \times 10^{-3}$ m, or 28 % of the average value estimated by the SEC method in September 2019 ($2.65 \times 10^{-2}$ m). The comparison improves when comparing the modelled $z_{0m}$ to VPEC measurements from September 2016 and August 2017 (T20), the model now overestimates the estimated $z_{0m}$ by in situ observations by $1.1 \times 10^{-3}$ m, or 9 % of the observed value ($1.25 \times 10^{-2}$ m).
As these data contain more wind directions, the overestimation of $z_{0m}$ in the southerly fetch directions is compensated by an underestimation in the easterly directions (Fig. 8). The difference between different in situ data highlights the variability in $z_{0m}$ in time, but also the uncertainty in the field measurements. The difference in averaged estimated $z_{0m}$ using in situ observations during the overlapping period across all wind directions is 12 % between the VPEC and the SEC methods.

The ice hummocks seen in the easterly directions have smaller $H$ and $\lambda$, which results in a smaller $z_{0m}$ than in the southerly directions. This is due to the anisotropic nature of the ice hummocks, and is confirmed by the eddy-covariance observations, regardless of the season. The extent of the UAV survey allows the application of the drag model for wind directions that rarely occur during the measurement period. This is particularly useful for the development of $z_{0m}$ parametrizations in atmospheric models. Interestingly, the topography at site S5 translates in a wavy pattern of $z_{0m}$ as function of wind direction, with two local minima at fetch directions of 90 and 180 degrees (Fig. 8).

To summarize, three independent but co-located methods, namely UAV photogrammetry, ICESat-2 laser altimetry and in situ eddy-covariance measurements, allow us to estimate the aerodynamic roughness of a rough ice surface at a specific site. The comparison confirms our two initial hypotheses: (1) the variability of estimated $z_{0m}$ using in situ observations as function of wind direction found by T20 is indeed a consequence of the anisotropic topography, and (2) the ICESat-2 data are very well suited to estimate $z_{0m}$ of a rough ice surface in both space and time. Without correcting for the residual scatter in photon elevations, the 1 m resolution ICESat-2 profiles most likely provide a lower bound of roughness, as they underestimate $z_{0m}$ by almost s factor 2 at the two rough ice locations in areas A and B. On the other hand, an attempt to account for this residual scatter may lead to an overestimated $z_{0m}$, by a factor that depends on the noise in the raw altimeter data. Nevertheless, given the fact that $z_{0m}$ varies over several orders of magnitude, we deem this method useful to understand the spatio-temporal variability of the aerodynamic roughness length over the GrIS.

## 4.4 Results: Mapping the roughness length $z_{0m}$ using ICESat-2

In this section we apply the elevation profile filtering described in Section 3 and the R92 model with parameterized $C_d$ (see Appendix A) to ICESat-2 ATL03 data to model and map the aerodynamic roughness ($z_{0m}$) over the K-transect. We process nearly two years of ICESat-2 ATL03 measurements, taken between 16 October 2018 and 06 September 2020. The results, without accounting for the unresolved photon scatter, are presented in Fig. 9. Within a distance of 10 km from the ice margin, $z_{0m}$ ranges between $10^{-3}$ m and $10^{-1}$ m. There is a clear transition of $z_{0m}$ values that separates the rough (S4, S5, KAN_L, SHR) and smooth (S6 and higher) surface in the ablation zone. Within a distance of several km, $z_{0m}$ can vary by more than one order of magnitude, e.g. north of locations S5 and KAN_L (Fig. 9). In order to quantify the variability in time, we group the $z_{0m}$ values from ICESat-2 in two groups, July – September and October – June, which corresponds to ICESat-2 cycles 1-3 & 5-8, and 4 & 8 respectively. The average $z_{0m}$ value for the two groups in each 200 m elevation bin is presented in Fig. 10. During summer, the average $z_{0m}$ value is $1 \times 10^{-2}$ m below 600 m asl, while it is around $6 \times 10^{-3}$ m during the other months. The average roughness approaches its minimum value of $10^{-4}$ m above 1000 m asl, regardless of the time period. When the ICESat-2 altimeter does not detect any obstacle, the bulk drag model only accounts for skin friction, which is prescribed as a constant in the model. Interestingly, $z_{0m}$ decreases very near the ice margin, which might be explained by the decreasing ice velocity at the margin, as most of the glaciers in this area are land-terminating.

The measurements described in Table 1 are also included in Fig. 10. The comparison indicates that the satellite product captures the overall variability along the K-transect (Fig. 10). Especially at lower elevations, the modelled $z_{0m}$ is within the range of in situ observations. The in situ roughness $z_{0m}$ can vary due to changing wind direction, but also due to instrumental

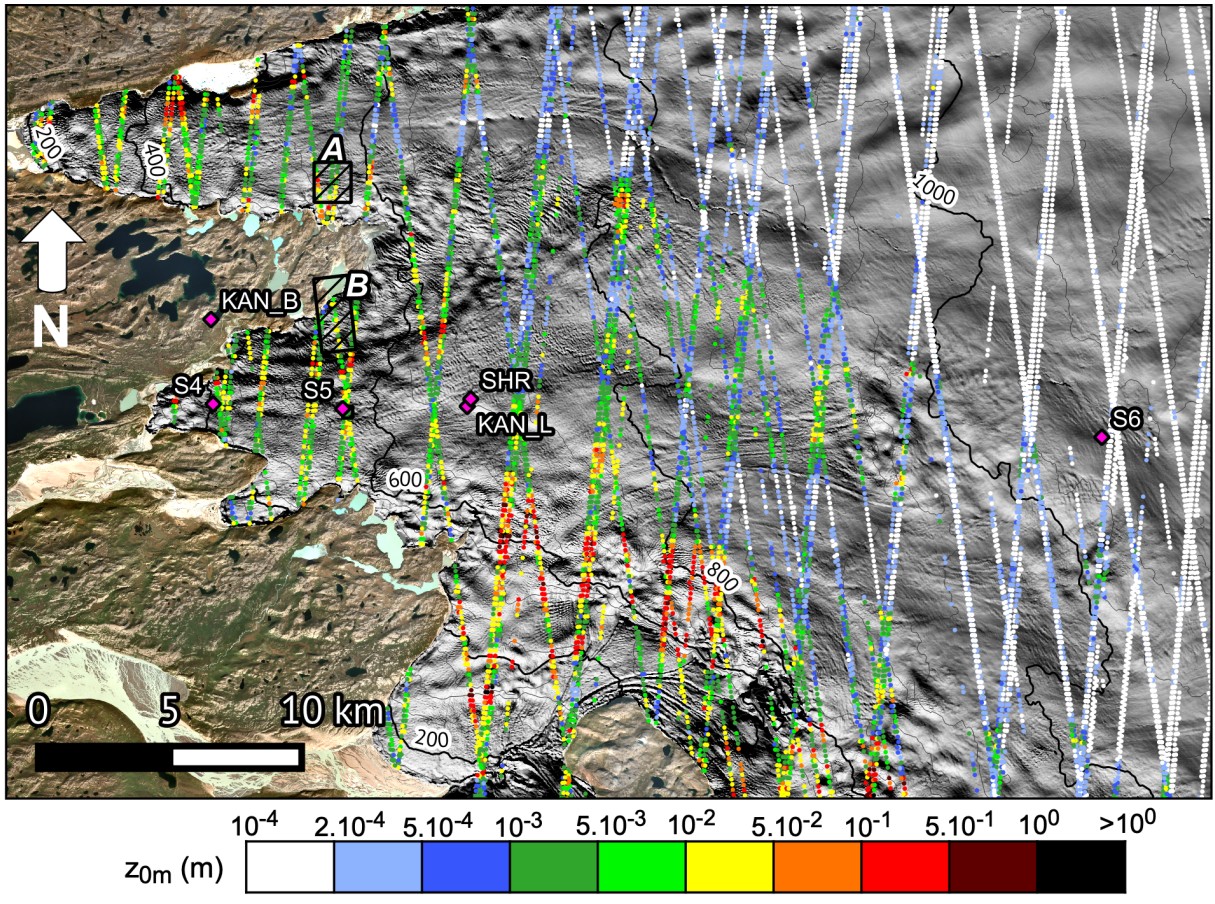

**Figure 9.** As in Fig. 6 but for the estimated aerodynamic roughness length ($z_{0m}$), without accounting for the residual photon backscatter.

uncertainty. Especially the smooth sites where the profile method has been used can exhibit large variability, such as at site S9 (see Table 1).

400     Unfortunately, the ICESat-2 altimeter is not able to detect obstacles that contribute to form drag at sites S6 (1010 m asl) and higher. At S6, the surface is flat during winter, but becomes rough during summer with ice hummocks with 0.6 m average height (SB08). Unfortunately the horizontal extent of these obstacles is smaller than the ICESat-2 footprint diameter ($\approx 15$ m). Higher up, the ice hummocks become even smaller and the surface eventually becomes snow-covered year-round. Nevertheless, snow sastrugi, known to reach up to 0.5 m height at site S10 from photographic evidence, still contribute to form drag. This results

405     in a maximum observed value of $z_{0m} = 7 \times 10^{-4}$ m at sites S9 and S10 (Fig. 10). Using a rough estimate for both $H$ and $\lambda$ at S6 and S10, based on photographs taken during the end of the ablation season, yields more realistic values for $z_{0m}$ (Fig. A1) than using $H$ and $\lambda$ from the ICESat-2 elevation profiles. Therefore we conclude that the roughness obstacles are not properly resolved at these locations in the ATL03 data using the algorithm presented in this study, even when the correction using the residual photons scatter is applied. This is mainly due to the limited footprint of the ICESat-2 measurements, but

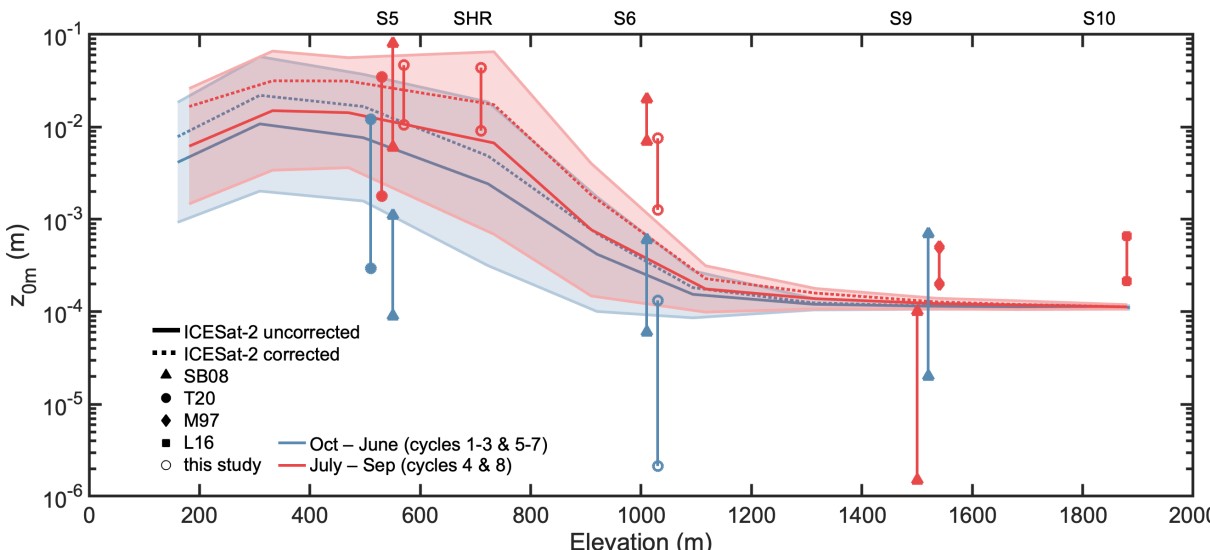

**Figure 10.** Estimated aerodynamic roughness length ($z_{0m}$) from ICESat-2 between 16 October 2018 until 06 September 2020 along the K-transect. The data were averaged over 200 m elevation bins and two time periods: summer (July – September) and winter (October – June). The variability range denotes the two-sided standard deviation within one elevation class. The in situ observations are described in Table 1.

also due to the orientation of the surface features, which limits the detectability of highly anisotropic features from 1D profile measurements. These limitations in the ICESat-2 measurements result in a uniform prescribed value of $z_{0m} = 1 \times 10^{-4}$ m for elevations above $\approx 1000$ m asl. The algorithm described in Sect. 3 could be adapted to extract these features from the ATL03 data. For instance, smaller-scale obstacles could be retrieved in multiple directions at cross-over points, using the information from multiple ICESat-2 tracks. However, this is beyond the scope of this study, which is to map the aerodynamic roughness of rough ice at large scales. For now, the implications of these findings for the sensible heat flux, and thus surface ablation, remain to be investigated. The areas with high $z_{0m}$ are in the low-lying ablation zone close to the ice edge, where the highest melt rates are observed. Accounting for the variable roughness might shed light on the drivers of these high melt rates.

## 5    Conclusions

The aerodynamic roughness of a surface ($z_{0m}$) in part defines the magnitude of the surface turbulent energy fluxes, yet is often poorly known for glaciers and ice sheets. We adapt the bulk drag partitioning model from Raupach (1992) such that it can be applied to 1D elevation profiles. Forcing this model with 1 m resolution elevation profiles taken from the ICESat-2 satellite laser altimeter, $z_{0m}$ becomes a quantifiable and mappable quantity. The model assumes that the surface is composed of regularly spaced, identical obstacles, which all have the same drag coefficient. Despite the fact that the drag coefficient for each individual obstacle remains poorly known, the evaluation in this study against different in situ observations, using different

techniques and for different locations and time periods, demonstrates the validity of this model. On the other hand, the use of the model of Lettau (1969) is not recommended over a rough ice surface, as it does not separate the form drag and the skin friction, and neglects both the effects of the displacement height and of inter-obstacle sheltering.

Mapping surface obstacles at 1 m resolution using the ICESat-2 altimeter data proves possible, as long as the roughness obstacles are large enough (e.g. crevasses, ice hummocks). Obstacles that are small compared to the ICESat-2 footprint diameter of $\approx 15$ m, such as ice hummocks found above 1000 m elevation in summer, or snow sastrugi expected year-round at even higher locations on the ice sheet from photographic evidence, are not resolved by the ICESat-2 measurements when used in combination with the methods presented in this study. This translates in a lower bound of $z_{0m} \approx 10^{-3}$ m that can realistically be mapped using this method. Furthermore, accounting for the scatter in the unresolved altimeter signal leads to overestimates of the aerodynamic roughness, as this scatter is a consequence of many different processes that must individually be modelled.

The methods presented in this paper can effectively be used to map $z_{0m}$ at ice sheet elevations below 1000 m. This lower ablation area is also where the contribution of turbulent heat fluxes to surface ablation, and thus runoff, is the largest. As a consequence of the orientation of the ICESat-2 orbit, the modelled $z_{0m}$ must be interpreted as the roughness that would be felt by air flowing in the direction parallel to each laser track. Surfaces of glaciers are often anisotropic, and $z_{0m}$ can vary by over one order of magnitude depending on the local wind direction.

The implications of the highly variable aerodynamic roughness for turbulent heat fluxes, and thus surface ablation, remains to be investigated. As current regional climate models typically use constant values for $z_{0m}$, these implications can be significant, especially in the lower ablation area where most of the surface runoff is generated. This analysis revealed for instance that highly crevassed areas have aerodynamic roughness values over $10^{-1}$ m, two orders of magnitude larger than typically used in regional climate models over bare ice.

*Data availability.* The data used in the figures can be downloaded here: https://doi.org/10.5281/zenodo.4386867. Additional data can be obtained from the authors without conditions.

## Appendix A:  Bulk drag model

For a single roughness element of height $H$ and frontal area $A_f$, placed on a horizontal area $A_l$, Raupach (1992, R92) models the form drag as :

$$\lim_{\lambda \to 0} \tau_r = \frac{F_D}{A_l} = \rho C_d \frac{A_f}{A_l} u(H)^2 = \rho C_d \lambda u(H)^2, \tag{A1}$$

where $F_d$ is the pressure drag force exerted on the obstacle, $H$ is the obstacle height, $\lambda$ the frontal area index and $C_d$ the drag coefficient of the obstacle. An important uncertainty resides in choosing an accurate value for $C_d$, due to its dependence on the shape of the obstacles, on the Reynolds number, and on the surface texture. Based on the analysis by Garbrecht et al. (2002)

for sea-ice pressure ridges, we choose the following parameterization,

$$C_d = \begin{cases} \dfrac{1}{2}(0.185 + 0.147H) & \text{if } H \le 2.5 \text{ m} \\ \dfrac{1}{2}\left(0.22\log(\dfrac{H}{0.2})\right) & \text{if } H > 2.5 \text{ m} \end{cases} \tag{A2}$$

Note that the factor $1/2$ is a consequence of a different definition for $C_d$ in Garbrecht et al. (2002) than Eq. (A1).

Similarly, R92 models the skin friction for an unobstructed flat surface as:

$$\lim_{\lambda \to 0} \tau_s = \rho C_s(z)u(z)^2 \tag{A3}$$

where $C_s(z)$ is the drag coefficient of the flat surface, referenced at a height $z$. Following Andreas (1995), $C_s(z)$ is estimated from the 10-m drag coefficient $C_s(10)$ measured over a flat surface, according to:

$$C_s(z) = \left[C_s(10)^{-0.5} - \frac{1}{\kappa}\left(\ln\left(\frac{10-d}{z-d}\right) - \widehat{\Psi_m}(z)\right)\right]^{-2}, \tag{A4}$$

with $C_s(10) = 1.2071 \times 10^{-3}$, which yields $z_{0m} = 10^{-4}$ m for a perfectly flat surface in this model. This value was chosen as it is the minimum value estimated using in situ observations by Smeets and Van den Broeke (2008) during winter over different snow surfaces.

In reality, the total surface shear stress is the sum of both the form drag on each individual obstacle ($\tau_r$), and the skin friction on the underlying surface ($\tau_s$) (Eq. (3)). Furthermore, an additional complexity arises at increasing obstacle frontal area index ($\lambda$), as the obstacles may effectively shelter a part of the surface and each other, thereby reducing both the skin friction and the form drag. Based on the previous work of Arya (1975), and on scaling arguments of the effective shelter volume, R92 includes sheltering and models the total surface shear stress over multiple obstacles as:

$$\tau(\lambda) = \tau_s(\lambda) + \tau_r(\lambda) \tag{A5}$$

$$= \rho u(H)^2 \left[C_s(H)\exp\left(-c\lambda\frac{u(H)}{u_*}\right) + \lambda C_d\exp\left(-c\lambda\frac{u(H)}{u_*}\right)\right], \tag{A6}$$

where $c = 0.25$ is an empirical constant that determines the sheltering efficiency. The latter equation may be written in the form:

$$Xe^{-X} = a, \tag{A7}$$

with:

$$X = \frac{c\lambda}{2}\frac{u(H)}{u_*}, \tag{A8}$$

$$a = \frac{c\lambda}{2}(C_s + \lambda C_d)^{-0.5}, \tag{A9}$$

which is solved iteratively using $X_{i+1} = e^{X_i}$ and $X_0 = a$, after R92. The solution yields $\dfrac{u(H)}{u_*}$.

The conversion of $\frac{u(H)}{u_*}$ to $z_{0m,R92}$ is finally possible using the semi-logarithmic wind profile Eq. (2) and referencing the wind speed at $z = H$. However, an expression for the displacement height $d$ and the roughness sublayer profile function $\widehat{\Psi_m(z)}$ is still required. For the displacement height, the simplified expression by Raupach (1994) is used:

$$d = H \left[ 1 - \frac{1 - \exp(-\sqrt{c_{d1}\lambda})}{\sqrt{c_{d1}\lambda}} \right] \tag{A10}$$

with $c_{d1} = 7.5$, which is then used to derive the value for $\Psi_r$ at height $z = H$ using the following expression:

$$\widehat{\Psi_m(H)} = \log(c_w) - 1 + c_w \tag{A11}$$

where,

$$c_w = \frac{z_* - d}{H - d}, \tag{A12}$$

in which $z_*$ is the upper height of the roughness layer. Raupach (1994) empirically determined that $c_w = 2$, which yields $\widehat{\Psi_m(H)} = 0.193$.

To summarize, the aerodynamic roughness length $z_{0m}$ of an elevation profile of length $L$ is modelled according to the following steps:

- The elevation profile is high-pass filtered using a cut-off wavelength $\Lambda = 35$ m.

- The obstacle height ($H$) is set to twice the standard deviation of the filtered profile.

- Each group of consecutive positive heights is defined as a single obstacle, which yields the amount of obstacles ($f$) per profile length ($L$).

- The frontal area index ($\lambda$) is calculated using Eq. (6).

- The displacement height $d$ is estimated with Eq. (A10).

- $C_s(z)$ is evaluated at $z = H$ using Eq. (A4), while $C_d$ is parameterized using Eq. (A2).

- $\frac{u(H)}{u_*}$ is estimated from Eqs. (A7) - (A9).

- $z_{0m}/H$ is estimated by evaluating the logarithmic wind profile at a height $z = H$, using Eq. (2)

Following the steps above, $z_{0m}$ can be estimated for any $H$ and $\lambda$, which is done in Fig. A1. At areas A, B and site S5, $H$ and $\lambda$ are estimated from the UAV surveys and from ICESat-2 data. At site S6, we assume that $H = 0.6 \pm 0.1$ m and $\lambda = 0.045 \pm 0.015$, based on photographs taken during the end of the ablation season. At the highest site S10, we assume that $H = 0.3 \pm 0.2$ m and $\lambda = 0.02 \pm 0.01$, which are typical values for sastrugi (Andreas, 1995).

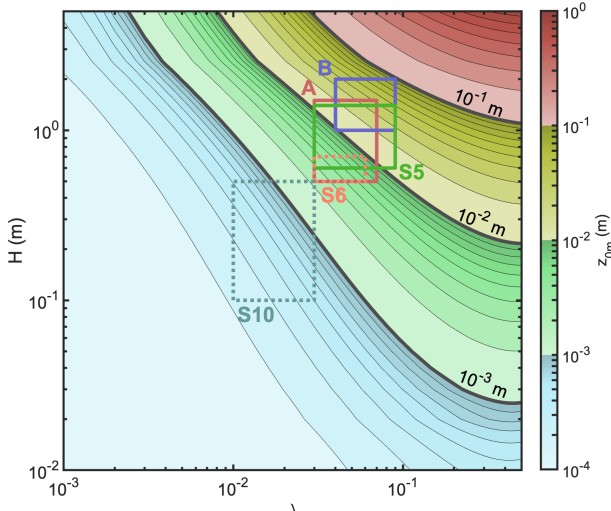

**Figure A1.** Estimated $z_{0m}$ using the R92 model with parameterized $C_d$ (Appendix A), as function of obstacle height $H$ and frontal area index $\lambda$. The solid squares denote the estimated $H$ and $\lambda$ at three sites using UAV surveys. The dashed squares are estimates based on photographs taken at the end of the ablation season. See Fig. 1 for the location of each site.

Other attempts have been made to relate $z_{0m}$ to the geometry of multiple surface roughness elements. For instance Lettau (1969, L69) empirically relate $z_{0m}$ to the average frontal area index of the roughness obstacles, which has been adapted by Munro (1989) for the surface of a glacier:

$$z_{0m,L69} = 2C_d H \frac{A_f}{A_l} = 2C_d H \lambda. \tag{A13}$$

Note that Eq. (A2) was adapted in order to be consistent with the definition of $C_d$ in Eq. (A1). In fact, Macdonald et al. (1998, M98) have shown that Eq. (A13) can be obtained by assuming that there is only form drag, and by setting $d = 0$, $\widehat{\Psi_m(z)} = 0$ and $C_d = 0.25$. By including the displacement height $d$, M98 is able to reproduce the non-linear feature of the $\frac{z_{0m}}{H} = f(\lambda)$ curve:

$$z_{0m,M98} = (H - d)\exp\left(-\left[\frac{C_d}{\kappa^2}\lambda\left(1 - \frac{d}{H}\right)\right]^{-0.5}\right). \tag{A14}$$

## Appendix B: Sensitivity experiments

### Cutoff wavelength $\Lambda$

We find that the optimal value of the cutoff wavelength for the high-pass filter is $\Lambda = 35$ m. This may be explained by the fact that the resulting filtered topography using $\Lambda = 35$m still contains most ($\approx 80$ %) of the total variance of the slope spectrum. The latter is defined as the power spectral density of the first derivative of the elevation profile. A sensitivity experiment using different values for $\Lambda$ at S5 can be found in Fig. B1. Changing the value for $\Lambda$ strongly impacts the estimated $H$ (Fig. B1c),

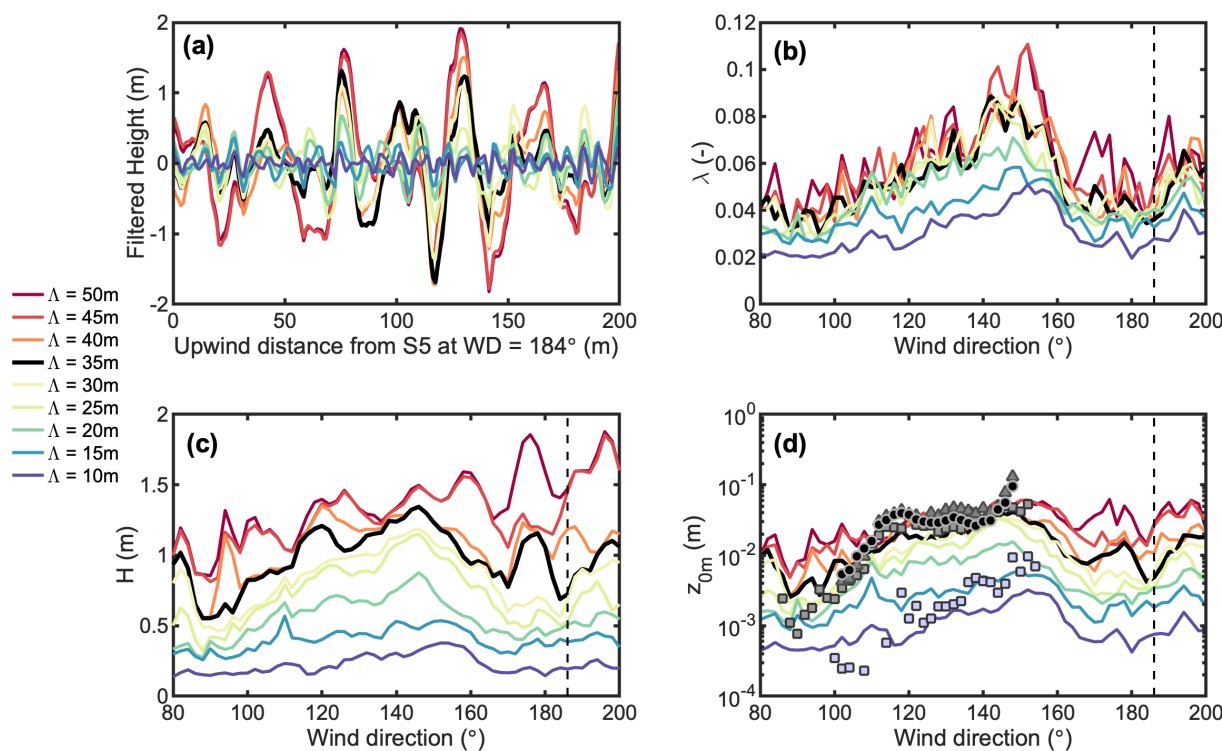

**Figure B1.** (a) Filtered elevation profile in direction $186°$, (b) estimated obstacle frontal area index, (c) estimated obstacle height and (d) modelled aerodynamic roughness length at site S5 for different high-pass cutoff wavelengths $\Lambda$. See Figure 8 in main text for the labels in panel d).

as the elevation profiles considered here contain information at all wavelengths (Fig. B1a). On the other hand, increasing the value for $\Lambda$ above 35m does not significantly affect the estimate frontal area index $\lambda$ (Fig. B1b). Overall, increasing $\Lambda$ from 10 m to 50 m increases the modelled $z_{0m}$ from $7.6 \times 10^{-4}$ m to $2.8 \times 10^{-2}$ m at S5, in the direction $184°$ that matches the ICESat-2 track (Fig. B1d).

**ATL03 kriging parameters**

In order to interpolate the geolocated photons product ATL03 in a regular 1-m resolution elevation profile, a fixed set of interpolation parameters was used, referred to as the default set. These are the median filter coefficients in Eq. (7) $q_{low} = 1$ and $q_{high} = 2$, the median filter window length of 50 m, the choice of a gaussian covariance function with a radius of 15 m in the kriging equations, and the maximum distance of photon distance to each regular grid point of 15 m.

This default parameter set was found to give robust results, even when only medium or low confidence photons are present in the ATL03 data. A sensitivity experiment by varying each parameter separately in a 200-m portion of areas A and B is given

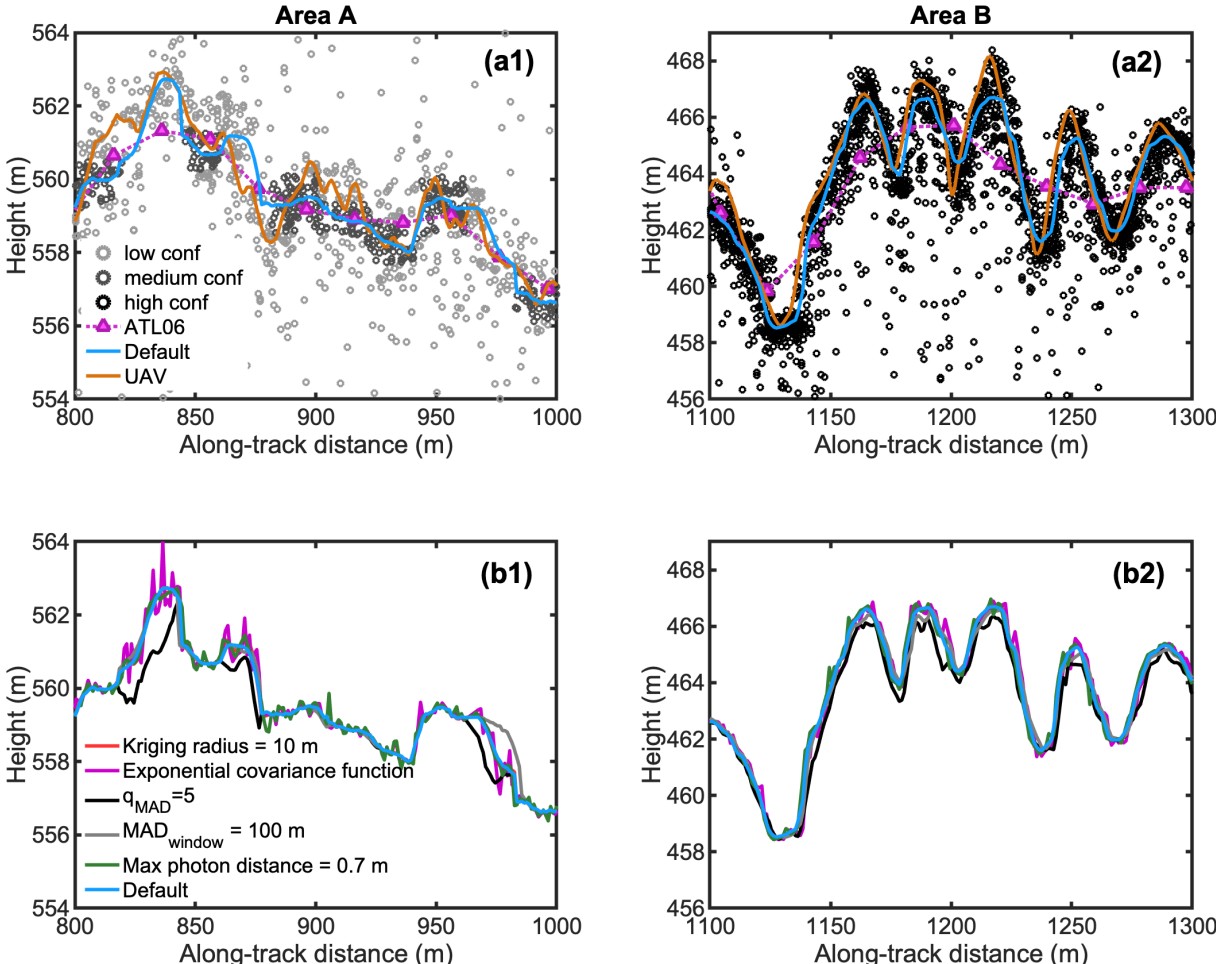

**Figure B2.** Elevation profiles in a 200-m portion of area A (left) and area B (right). The top panels contain the ATL03 data sorted in confidence levels (dots), the ATL06 data (pink triangles), the profiles measured by UAV photogrammetry (orange line) and the 1-m interpolated ATL03 data using the default settings used in the main text (blue line). The bottom panels contain the 1-m interpolated AT03 data using different origins and photon filtering settings.

in Fig. B2. While the interpolated ATL03 elevation still misses small-scale features present in the UAV data, varying each parameter does not give improved results (Fig B2).

*Author contributions.* PS, MRvB, CHR and WI secured funding for this study and for the field experiments. JFS piloted the UAV in the field, PS and MvT acquired and processed the eddy-covariance data. EJN, JFS and WI made the UAV elevation models. MvT and BW processed the ICESat-2 data. MvT, with the help of all the co-authors, designed the study and wrote the first draft. All authors participated in the review and in the editing of the manuscript.

*Competing interests.* The authors declare that they have no conflict of interest.

*Acknowledgements.* The authors thank all the people and institutes that help maintaining the instruments in the field. We are grateful to Nanna Karlsson, Dirk van As and Giorgio Cover for their support in the field. This work is funded by the Utrecht University and by the Netherlands Polar Program (NPP), of the Netherlands Organisation of Scientific Research, section Earth and Life Sciences (NWO/ALWOP.431). This work was carried out on the Dutch national e-infrastructure with the support of SURF Cooperative. JFS and WI acknowledge support by the Netherlands Organization for Scientific Research NWO [016.181.308] and European Research Council [676819]. The views and interpretations in this publication are those of the authors and are not necessarily attributable to ICIMOD.

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
