# Peer review of "Mapping the aerodynamic roughness of the Greenland Ice Sheet surface using ICESat-2: Evaluation over the K-transect"

_The Cryosphere, 2020_

## Referee Comment (RC1)

**Review of the paper: Mapping the aerodynamic roughness of the Greenland ice sheet surface using ICESat-2: Evaluation over the K-transect**
by Maurice van Tiggelen and others

A bulk drag parameterizatzion is applied to calculate the aerodynamic roughness length over a part of the western Greenland ice sheet as a function of the surface topography that has been evaluated using UAV photogrammetry and finally ICESat-2 laser altimeter measurements. The parameterization includes skin drag and form drag caused by small scale features such as hummocks and sastrugi. Results for the roughness are compared with those obtained from in situ turbulence measurements. Finally, a map of the surface roughness is presented over a selected region of the western ice sheet.

In most parts the paper is very well written and it follows a clear logic presenting novel results. Results might become helpful to better understand in the future the role of surface roughness for atmospheric and ice processes. I suggest, however, an improvement of the description of the used roughness parameterization before publication.

**Major Revisions**

1. Please separate more clearly in 2.1 the description of the determination of $z_{0m}$ from the measured fluxes and from the used model. Perhaps, introduce corresponding headings so that the structure becomes clear at a first glance.

2. It seems that a mixture is used here of the schemes by R92, Andreas (1995) and of own assumptions. E.g., equation (A3) ignores the wake effect. Please compare this with equation (7) of Andreas (1995). This needs explanation. Please clearly specify own assumptions.

3. In its present version equation (A4) is wrong. This can be seen by inserting the value $z = 10$ m. Probably, a missprint (?)

4. I understood that $\widehat{\psi_m}$ is set to zero to derive $z_{0m}$ from measurements. But this differs from the assumptions in the Appendix for the most complex scheme. Please better explain why this is no contradiction.

5. I propose to describe in the Appendix first the complete scheme by R92 (in its version used here), and then give equations (A5) and (A6) of others. This would facilitate reading.

6. The obstacle height is set twice the standard deviation of the filtered profile. How sensitive are the results to this assumption?

7. Equation (A2) (upper line) has been given in Garbrecht et al. (2002) (not Garbrecht et al. (1999) as in the lower line).

8. Line 80: Equation (3) is used by Lüpkes et al. (2012) and by Lüpkes and Gryanik (2015) as well. The difference is that the width of the roughness elements (ice floes) can be of the same order as the width of open water fetch. However, exactly the same equation (3) is used by Garbrecht et al. (1999, 2002) and by Castellani et al. (2014), who parameterize the impact of ridges on sea ice. The difference in their models to the one discussed in the manuscript is that due to the large distances between ridges further simplifications are possible.

9. Figure 6: It should be mentioned that the 'observed' $z_{0m}$ depends also on a model, namely on all assumptions involved in equation (2) when it is applied over inhomogeneous surface topography. This would be different if just drag coefficients were compared with each other, for which just the observed wind speed and momentum fluxes at the measurement height would be needed.

**Minor revisions**

1. Line 32: here it might be useful to cite cite also Lüpkes and Gryanik (2015).

2. Line 36: perhaps after 'the application of such models' in weather and climate models.

3. Section 2.1, the hat over $\psi_m$ should always appear as in equatuion (1).

4. Figure 6, caption: The solid grey symbols are not really measurements of $z_0$. These points have probably been derived from wind and flux measurements applying equation (2). That's a large difference because equation (2) is also a kind of model. Please, add also equation numbers for the different $z_{0m}$ data.

5. Line 273: one could add here that also Lüpkes et al. (2012) use constant $C_d$ (which is $c_w$ in their paper).

6. line 315: compare $H$ and $\lambda$ .... you mean: compare with satellite and UAV measurements?

7. Figure 8: I do not understand the shift of the orange dotted line. Perhaps I have overseen the explanation? Also in the caption, which modelled $z_{0m}$? There are several approaches....

8. line 334: 'between different in situ' ? Forgotten data?

9. line 337: better write somethink like: hummocks having been formed during westerly wind have usually ....

**Reference:**

Garbrecht T., Lüpkes C., Hartmann J., Wolff M., (2002) Atmospheric drag coefficients over sea ice - validation of a parameterisation concept, Tellus, 54A, 205-219.

Castellani, G., Lüpkes, C., Hendricks, S., Gerdes, R. (2014) Variability of Arctic sea ice topography and its impact on the atmospheric surface drag, J. Geophs. Res. Oceans, 119, doi:10.1002/2013JC009712.

Lüpkes, C. and V.M. Gryanik (2015), A stability dependent parametrizationn of transfer coefficients for momentum and heat over polar sea ice to be used in climate models, J. Gephys. Res. Atmos, 120, doi: 10.1002/2014JD022418

---

## Author Comment (AC2)

**Response to referee 1**

A bulk drag parameterizatzion is applied to calculate the aerodynamic roughness length over a part of the western Greenland ice sheet as a function of the surface topography that has been evaluated using UAV photogrammetry and finally ICESat-2 laser altimeter measurements. The parameterization includes skin drag and form drag caused by small scale features such as hummocks and sastrugi. Results for the roughness are compared with those obtained from in situ turbulence measurements. Finally, a map of the surface roughness is presented over a selected region of the western ice sheet. In most parts the paper is very well written and it follows a clear logic presenting novel results. Results might become helpful to better understand in the future the role of surface roughness for atmospheric and ice processes. I suggest, however, an improvement of the description of the used roughness parameterization before publication.

We are grateful to the referee for his thoughtful and precise comments. In the text below we respond point-by-point and discuss the changes to the work. The referee comments are written in black. Our answer to each comment is written below in blue. The proposed changes to the original manuscript are then highlighted in red.

**Major Revisions**

1. Please separate more clearly in 2.1 the description of the determination of $z_{0m}$ from the measured fluxes and from the used model. Perhaps, introduce corresponding headings so that the structure becomes clear at a first glance.

We agree with the reviewer and therefore propose to add a third sub-section to separate the definition of $z_{0m}$ from the bulk model for $z_{0m}$.

L60:

2 Model

2.1 Definition of the aerodynamic roughness length $z_{0m}$
(...)
Hence, the process of finding $z_{0m}$ is equivalent to finding $d$, $\dfrac{u(z)}{u_*}$ and $\widehat{\Psi_m(z)}$ simultaneously.

2.2 Bulk drag model of $z_{0m}$

The main task is to model the total surface shear stress $\tau = \rho u_*^2$, which for a rough surface is the sum of both form drag $\tau_r$ and skin friction $\tau_s$:
(...)

2.3 Definition of the height (H) and frontal area index ($\lambda$) over a rough ice surface
(...)

2. It seems that a mixture is used here of the schemes by R92, Andreas (1995) and of own assumptions. E.g., equation (A3) ignores the wake effect. Please compare this with equation (7) of Andreas (1995). This needs explanation. Please clearly specify own assumptions.

The reviewer is right, we have applied the R92 model to a realistic surface (rough ice), just as Andreas (1995) did for sastrugi. Our equation (A3) is in fact Equation (12) from R92, which aims to model the skin friction for a flat surface without any obstruction by roughness elements. However in this study we do take into account wake effects that occur when $\lambda > 0$ as in Andreas (1995), in our equation (A7). We have added clarification with our Equations (A3) and (A7) and hope this becomes more clear in the revised manuscript.

L428:
Similarly, R92 models the skin friction for an unobstructed flat surface as:

$$\lim_{\lambda \to 0} \tau_s = \rho C_s(z)u(z)^2 \tag{A3}$$

L445:
Based on the previous work of Arya (1975), and on scaling arguments of the effective shelter volume, R92 includes sheltering and models the total surface shear stress over multiple obstacles as:

$$\tau(\lambda) = \tau_s(\lambda) + \tau_r(\lambda) \tag{A7}$$
$$= \rho u(H)^2 \left[ C_s(H)\exp\left(-c\lambda\frac{u(H)}{u_*}\right) + \lambda C_d\exp\left(-c\lambda\frac{u(H)}{u_*}\right) \right],$$

where $c = 0.25$ is an empirical constant that determines the sheltering efficiency.

3. In its present version equation (A4) is wrong. This can be seen by inserting the value z = 10 m. Probably, a missprint (?)

The reviewer is correct, there are two misprints in our equation (A4). The height of the obstacles $H$ should be replaced by the variable $z$. A minus sign was also missing in the exponent of $C_s(10)$. The correct version of the equation was used in the code, therefore this does not affect the results in any way. We have corrected these misprints in the revised version.
L429
Following Andreas (1995), $C_s(z)$ is estimated from the 10-m drag coefficient

$C_s(10)$ measured over a flat surface, according to:

$$C_s(z) = \left[ C_s(10)^{-0.5} - \frac{1}{\kappa} \left( \ln\left( \frac{10-d}{z-d} \right) - \widehat{\Psi_m}(z) \right) \right]^{-2}, \qquad \text{(A4)}$$

with $C_s(10) = 1.2071 \times 10^{-3}$, which yields $z_{0m} = 10^{-4}$ m for a perfectly flat surface in this model.

4. I understood that $\widehat{\Psi_m(z)}$ is set to zero to derive $z_{0m}$ from measurements. But this differs from the assumptions in the Appendix for the most complex scheme. Please better explain why this is no contradiction.

Using the bulk drag model of R92, the estimation of $z_{0m}$ requires modelling the drag coefficient, thus the wind speed and the momentum flux, at the top of the roughness elements $(z = H)$. At this height the averaged vertical profiles of horizontal wind velocity deviate from the inertial sublayer wind profile by an offset $\widehat{\Psi_m(z)}$. However for $z > 2H$ we assume that the inertial sublayer profile is valid again, and defined by a roughness length $z_{0m}$. Thus, linking the $z_{0m}$ that defines the wind profile in the inertial sublayer $(z > 2H)$ to the wind speed at the top of the roughness elements $(z = H)$ requires correcting for the wind profile deviation by $\widehat{\Psi_m(z)}$. On the other hand, when estimating $z_{0m}$ from the measured wind speed and momentum flux, we assume that the instruments are located in the inertial sublayer where $\widehat{\Psi_m(z)} = 0$. This is most likely valid, given that we measure at $z = 3.7$ m and that $H < 1.5$ m at measurement site S5. We propose to add an explanatory sentence in our section *3.1 Eddy covariance measurements*:

L156

We only select data taken during near-neutral conditions $(z/L_o < 0.1)$, and we assume that the measurements are taken above the roughness layer, i.e. $\widehat{\Psi_m(z)}$ = 0. The latter is a reasonable assumption, given that the height of the obstacles $(H)$ at these sites is less than 1.5 m, which means that the roughness layer unlikely exceeds 3 m (Smeets et al., 1999; Harman and Finnigan, 2007). On the other hand, when applying the drag model to estimate $z_{0m}$ (Appendix A.), the correction factor $\widehat{\Psi_m(z)}$ is taken into account. The reason is that the obstacles are located in the roughness layer, where the vertical wind profiles deviate from the inertial sublayer wind profiles, according to Eq. (1).

5. I propose to describe in the Appendix first the complete scheme by R92 (in its version used here), and then give equations (A5) and (A6) of others. This would facilitate reading.

We agree with the reviewer, and we have moved equations (A5) and (A6) to the end of Appendix A.

L467:

Other attempts have been made to relate $z_{0m}$ to the geometry of multiple surface roughness elements. For instance Lettau (1969, L69) empirically relates $z_{0m}$ to the average frontal area index of the roughness obstacles, which has been adapted by Munro (1989) for the surface of a glacier:

$$z_{0m,L69} = 2C_d H \frac{A_f}{A_l} = 2C_d H \lambda. \tag{A13}$$

Macdonald et al. (1998,M98) have shown that Eq. (A13) can be obtained by assuming that there is only form drag, and by setting $d = 0$, $\widehat{\Psi_m(z)} = 0$ and $C_d = 0.25$. By including the displacement height $d$, M98 is able to reproduce the non-linear feature of the $\frac{z_{0m}}{H} = f(\lambda)$ curve:

$$z_{0m,M98} = (H - d)\exp\left(-\left[\frac{C_d}{\kappa^2}\lambda\left(1 - \frac{d}{H}\right)\right]^{-0.5}\right). \tag{A14}$$

[*end of Appendix A*]

6. The obstacle height is set twice the standard deviation of the filtered profile. How sensitive are the results to this assumption?

The elevation profiles we consider contain information at all wavelengths. Therefore, changing the value of the high-pass cutoff wavelength affects the resulting standard deviation, and thus the modelled value for $z_{0m}$. We propose to add a sensitivity analysis on the modelled $H$, $\lambda$ and $z_{0m}$ for $\Lambda \in [10; 50]$ m at site S5 in the new Appendix B (see Fig B1). We also propose to add a few explanatory sentences in the Appendix regarding this sensitivity.

L113:

To remove the influence of the widest obstacles, the elevation profile of length L is linearly detrended and the power spectral density of the detrended profile is computed in order to filter out all the wavelengths larger than the cutoff wavelength $\Lambda = 35$ m. This value is found to give optimal results, which is shown in Appendix B.

**Appendix B: Sensitivity experiments:**

**Cutoff wavelength $\Lambda$**

We find that the optimal value of the cutoff wavelength for the high-pass filter is $\Lambda = 35$ m. This may be explained by the fact that the resulting filtered topography using $\Lambda = 35$m still contains most ($\approx 80$ %) of the total variance of the slope spectrum. The latter is defined as the power spectral density of the first derivative of the elevation profile. A sensitivity experiment using different values for $\Lambda$ at S5 can be found in Fig. B1. Changing the value for $\Lambda$ strongly impacts the estimated $H$ (Fig. B1c), as the elevation profiles considered here contain information at all wavelengths (Fig. B1a). On the other hand, increasing the value for $\Lambda$ above 35m does not significantly affect the estimate frontal

[Figure]

Figure B1: (a) Filtered elevation profile in wind fetch direction 186°, (b) estimated obstacle frontal aera index, (c) estimated obstacle height and (d) modelled aerodynamic roughness length at site S5 for different high-pass cutoff wavelengths Λ. See Figure 8 in main text for the labels in d).

area index $\lambda$ (Fig. B1b). Overall, increasing $\Lambda$ from 10 m to 50 m increases the modelled $z_{0m}$ from $7.6 \times 10^{-4}$ m to $2.8 \times 10^{-2}$ m at S5, in the wind fetch direction 184° that matches the ICESat-2 track (Fig. B1d).

7. Equation (A2) (upper line) has been given in Garbrecht et al. (2002) (not Garbrecht et al. (1999) as in the lower line).
The reviewer is right. We have modified the reference accordingly.

L423:
Based on the analysis by Garbrecht et al. (2002) for sea-ice pressure ridges, we choose the following parameterization,

$$C_d = \begin{cases} \dfrac{1}{2}(0.185 + 0.147H) & \text{if } H \leq 2.5 \text{ m} \\ \dfrac{1}{2}\left(0.22\log(\dfrac{H}{0.2})\right) & \text{if } H > 2.5 \text{ m} \end{cases} \tag{A2}$$

Note that the factor $1/2$ is a consequence of a different definition for $C_d$ in Garbrecht et al. (2002) than Eq. (A1).

8. Line 80: Equation (3) is used by Lüpkes et al. (2012) and by Lüpkes and Gryanik (2015) as well. The difference is that the width of the roughness elements (ice floes) can be of the same order as the width of open water fetch. However, exactly the same equation (3) is used by Garbrecht et al. (1999, 2002) and by Castellani et al. (2014), who parameterize the impact of ridges on sea ice. The difference in their models to the one discussed in the manuscript is that due to the large distances between ridges further simplifications are possible.
We thank the reviewer for this clarification. We believe this information might be useful for the interested reader, and thus we propose the following modification :

L84:
At this point, we will differ from the model by Shao and Yang (2008), who add an extra term in Eq. (3) in order to separate the skin friction at the roughness elements and the underlying surface. We also differ from the models by Lüpkes et al. (2012) and Lüpkes and Gryanik (2015), where skin friction over sea-ice is separated between a component over open water, and a component over ice floes. In the case of a rough ice surface, their is no clear distinction between the obstacles and the underlying surface. Therefore, we follow the model of Raupach (1992, R92), which is designed for surfaces with a moderate frontal area index ($\lambda < 0.2$).

9. Figure 6: It should be mentioned that the 'observed' $z_{0m}$ depends also on a model, namely on all assumptions involved in equation (2) when it is applied over inhomogeneous surface topography. This would be different if just drag coefficients were compared with each other, for which just the observed wind speed and momentum fluxes at the measurement height would be needed.
We agree with the reviewer. We propose to replace the "measured $z_{0m}$" by "estimated $z_{0m}$ from in situ observations" everywhere in the text and in figure

captions.

**Minor Revisions**

1. Line 32: here it might be useful to cite cite also Lüpkes and Gryanik (2015).
Added

L32
Lüpkes et al. (2012) and Lüpkes & Gryanik (2015) developed a bulk drag model for sea-ice that is used in multiple atmospheric models.

2. Line 36: perhaps after 'the application of such models' in weather and climate models.
Changed

L36
The second challenge is the application of such models in weather and climate models, which requires mapping small-scale obstacles over large areas, e.g. an entire glacier or ice sheet.

3. Section 2.1, the hat over $\Psi_m$ should always appear as in equatuion (1).
We have chosen to use the notation from Harman and Finnigan (2007), where the hat notation is used for roughness sublayer variables. Therefore $\Psi_m$ and $\widehat{\Psi_m(z)}$ are two distinct quantities. We propose an extra sentence in Section 2.1 for clarification.

L71
The dependency of the eddy diffusivity for momentum on the diabatic stability and on the turbulent wake diffusion are described as $\Psi_m \left( \dfrac{z-d}{L_o} \right)$ and $\widehat{\Psi_m(z)}$, respectively, where $L_o$ is the Obukhov length. The hat notation is used for the roughness layer quantities, as in Harman and Finnigan (2007).

4. Figure 6, caption: The solid grey symbols are not really measurements of z0. These points have probably been derived from wind and flux measurements applying equation (2). That's a large difference because equation (2) is also a kind of model. Please, add also equation numbers for the different $z_{0m}$ data.
In accordance with previous Major comment #9, we propose to replace all the "measured $z_{0m}$" by "estimated $z_{0m}$ from in situ observations".

5. Line 273: one could add here that also Lüpkes et al. (2012) use constant Cd (which is cw in their paper).
added

The parametrization for Cd from Garbrecht et al. (1999) (Eq. (A2)), for which Cd increases with H, yields most acceptable results when used in combination with the R92 model (Fig. 6). Note that Lüpkes et al. (2012) use a constant value for $C_d$.

 6. line 315: compare H and $\lambda$ .... you mean: compare with satellite and UAV measurements?
Yes. We have modified the sentence for clarification.

L315
Although the UAV profile is too short to statistically compare H and $\lambda$ to the ICESat-2 altimeter, the qualitative comparison between the two confirms that the satellite altimeter is very well capable of detecting most of the obstacles that are smaller than 20 m in width.
7. Figure 8: I do not understand the shift of the orange dotted line. Perhaps I have overseen the explanation? Also in the caption, which modelled $z_{0m}$? There are several approaches....
The orange dotted line is the orange line divided by 10, and is therefore a crude guess of what the modelled $z_{0m}$ using UAV data would look like at site S5 in March. We propose to add an explanatory sentence. We also detailed which model was used in the caption. Note that we have also separated Fig. 8 in two parts, after a suggestion by Referee #3.

L319
Both H and $\lambda$ are smaller in the satellite profile than in the UAV profile, but the modelled $z_{0m}$ agrees qualitatively with the $z_{0m}$ estimated from AWS S5 measurements during March-April. During this time period, $z_{0m}$ is approximately a factor 10 smaller than during the end of the ablation season (Fig. 8, dashed orange line).

8. line 334: 'between different in situ' ? Forgotten data?
Corrected

L334:
The difference between different in situ data highlights the variability in $z_{0m}$ in time, but also the uncertainty in the field measurements.

9. line 337: better write somethink like: hummocks having been formed during westerly wind have usually ....
We do not discuss how the ice hummocks have been formed, which is outside the scope of this paper. Nevertheless the surface at S5 may be considered as homogeneously covered by nearly identical yet anisotropic ice hummocks, that have different heights and frontal area indices depending on the looking direction. We

[Figure]

Figure 8: (a) Drag model evaluation at site S5. (b): Drag coefficient for form drag ($C_d$) used in the bulk drag model (black line) or required to perfectly fit the observations. The orange solid line is the modelled $z_{0m}$ using the R92 model and UAV photogrammetry on 06 September 2019, while the dashed orange line is the orange line shifted down by a factor 10. Solid symbols are measurements from sonic eddy-covariance (SEC) or vertical propeller eddy-covariance (VPEC). Additional data is from van Tiggelen et al. (2020, T20). The vertical dashed line denotes the direction sampled by the ICESat-2 laser beam on 14 March 2019. The errorbar denotes the range between the uncorrected and corrected ICESat-2 measurements.

propose some minor changes in the revised manuscript for clarification. We also propose to update "westerly wind direction" in "easterly wind fetch direction".

L337:
The ice hummocks seen in the easterly wind fetch directions have smaller H and $\lambda$, which results in a smaller $z_{0m}$ than the hummocks seen in the southerly wind fetch directions. This is due to the anisotropic nature of the ice hummocks.

---

## Author Comment (AC3)

**Response to referee 2**

We thank the anonymous referee for her/his comments. In the text below we respond point-by-point and discuss the changes to the work. The referee comments are written in black. Our answer to each comment is written below in blue. The proposed changes to the original manuscript are then highlighted in red.

**General comments**

This is strong manuscript that demonstrates impressive proficiency with many different sources of data (AWS, UAV, ICESat-2, modeling). The methods are generally well- described. The results section is very interesting and the development of spatially extensive aerodynamic roughness lengths for the K-Transect from ICESat-2 is commendable.

However, I do recommend some revisions. In its current form, the introduction is poor. Some of the terminology is vague, references are lacking and the overall research is poorly motivated. I encourage the authors to revise it thoroughly and have provided some ideas for doing so below.
We thank the reviewer for this feedback. We agree that the research could benefit by an improvement of the Introduction, and we have thus adapted parts of it and included more references. We hope that the updated introduction better motivates this study.

While it is useful to know that the commonly used method for deriving z0m from ICESat-2 (i.e. the standard deviation of ATL03 heights) tends to overestimate z0m, the new measure is slightly unsatisfactory if it underestimates z0m by a factor two. Without looking at the data, it is difficult to discern why. It could be due to the slightly arbitrary choice of filtering (qlow = 1 and qhigh = 2) to remove photons above and below the median. It could due to the choice gaussian covariance function, window size or assumed wavelength. Given that this is one of the first papers to investigate roughness lengths using ICESat-2 and availability of ground-truth data, it would be useful if the authors could develop a more unbiased method. I would encourage the authors to perform some sensitivity tests with these choices to see if they would reduce bias in their ICESat-2 z0m products.
We thank the reviewer for pointing out a very important issue that this study leaves unsolved : the systematic underestimation of $z_{0m}$ when using the ICESat-2 measurements. Although we are also convinced that the current methods could be improved further, this would (given the current data) require an arbitrary tuning of the methods to fit the few available in situ observations. The arbitrary choices made in this study, such as the filter wavelength of 35 m, the median filter coefficients qlow = 1 and qhigh = 2, or the window size of 50 m, are unfortunately necessary in order to convert the raw photons to the final map of $z_{0m}$.

Nevertheless, we show that our results capture the observations well given the many uncertainties. Given that $z_{0m}$ is often taken constant or used as a tuning parameter in atmospheric models, we consider them as very useful. Besides, our aim is to lay a foundation for more sophisticated studies. Furthermore, the high spatial variability of $z_{0m}$ is a new result that has never been achieved using conventional in situ measurements. Finally, we would like to point out that $z_{0m}$ ranges over nearly 4 orders of magnitudes over the Greenland Ice Sheet, and that it is the natural logarithm of $z_{0m}$ that is used in atmospheric models to compute drag (our Eq. (1)). Therefore, we expect the 40% underestimation of $z_{0m}$ that we have found in area A to have a limited impact on momentum drag and turbulent fluxes.

In order to give the interested reader the required information to improve our methods, we propose to add a sensitivity analysis in the Appendix.
In our new Fig. B1 we illustrate the impact of different filter wavelength $\Lambda$ on the modelled $z_{0m}$ at site S5. Our chosen value of 35 m gives the most acceptable results compared to the AWS observations.
In our new Fig. B2 we compare the interpolated elevation profiles from ICESat-2 ATL03 data using different covariance functions, different kriging radii different nearest neighbour ranges, and different median filter parameters, over two 200 m profiles in areas A and B. Changing these parameters does not lead to a clear improvement in elevation profiles.

**Appendix B: Sensitivity experiments**

**Cutoff wavelength $\Lambda$**

We find that the optimal value of the cutoff wavelength for the high-pass filter is $\Lambda = 35$ m. This may be explained by the fact that the resulting filtered topography using $\Lambda = 35$m still contains most ($\approx 80$ %) of the total variance of the slope spectrum. The latter is defined as the power spectral density of the first derivative of the elevation profile. A sensitivity experiment using different values for $\Lambda$ at S5 can be found in Fig. B1. Changing the value for $\Lambda$ strongly impacts the estimated $H$ (Fig. B1c), as the elevation profiles considered here contain information at all wavelengths (Fig. B1a). On the other hand, increasing the value for $\Lambda$ above 35m does not significantly affect the estimate frontal area index $\lambda$ (Fig. B1b). Overall, increasing $\Lambda$ from 10 m to 50 m increases the modelled $z_{0m}$ from $7.6 \times 10^{4}$ m to $2.8 \times 10^{-2}$ m at S5, in the direction $184°$ that matches the ICESat-2 track (Fig. B1d).

**ATL03 kriging parameters**

In order to interpolate the geolocated photons product ATL03 in a regular 1-m resolution elevation profile, a fixed set of interpolation parameters was used, referred to as the default set. These are the median filter coefficients in Eq. (7) $q_{low} = 1$ and $q_{high} = 2$, the median filter window length of 50 m, the choice of a gaussian covariance function with a radius of 15 m in the kriging equations,

[Figure]

Figure B1: (a) Filtered elevation profile in direction 186°, (b) estimated obstacle frontal aera index, (c) estimated obstacle height and (d) modelled aerodynamic roughness length at site S5 for different high-pass cutoff wavelengths Λ. See Figure 8 in main text for the labels in d).

and the maximum distance of photon distance to each regular grid point of 15 m.

This default parameter set was found to give robust results, even when only medium or low confidence photons are present in the ATL03 data. A sensitivity experiment by varying each parameter separately in a 200-m portion of areas A and B is given in Fig. B2. While the interpolated ATL03 elevation still misses small-scale features present in the UAV data, varying each parameter does not give improved results (Fig. B2).

**Specific comments**

L16: Please consider capitalizing "ice sheet". It's the Amazon River, the Tibetan Plateau and should be the Greenland Ice Sheet. Indeed the Nature paper that you cite (Shepherd et al., 2020) has it this way.

The reviewer is correct. We have changed this accordingly, and we propose to use the acronym GrIS everywhere below L16, except in figure captions. The title of the manuscript was also corrected.

Title:

Mapping the aerodynamic roughness of the Greenland Ice Sheet surface using ICESat-2: Evaluation over the K-transect

L6:

[Figure]

Figure B2: Elevation profiles in a 200-m portion of area A (left) and area B (right). The top panels contain the ATL03 data sorted in confidence levels (dots), the ATL06 data (pink triangles), the profiles measured by UAV photogrammetry (orange line) and the 1-m interpolated ATL03 data using the default settings used in the main text (blue line). The bottom panels contain the 1-m interpolated AT03 data using different origins and photon filtering settings.

We apply the model to a rough ice surface on the K-transect (western Greenland Ice Sheet) using UAV photogrammetry, (...)
L16:
Between 1992 and 2018, the mass loss of the Greenland Ice Sheet (GrIS) contributed (...)
L18:
Runoff occurs mostly in the low-lying ablation area of the GrIS, where (...)
L50: (...) profiles measured over the west GrIS by the ICESat-2 laser altimeter.
Figure 1:
(c) Location of the K-transect on the Greenland Ice Sheet.
L145:
(...) mass balance observations on the western part of the GrIS (...)
Figure 5:
(...) lower part of the K-transect, West Greenland Ice Sheet.

L351:
(...) spatio-temporal variability of the aerodynamic roughness length over the GrIS.

L19: If you define an acronym, it is usually appropriate to use it here and elsewhere (e.g. L50, L146).

We have replaced Greenland Ice Sheet by GrIS in the remainder of the manuscript (see reply above).

L18-21: Please provide some references for these two statements. A lot of work has been done on these topics and it is negligent to overlook it.

We agree with the reviewer. We propose to add the following references in this paragraph:

L18:

Runoff occurs mostly in the low-lying ablation area of the GrIS, where bare ice is exposed to on-average positive air temperatures throughout summer (e.g. Smeets et al, 2018; Fausto et al, 2021). As a consequence, the downward turbulent mixing of warmer air towards the bare ice, the sensible heat flux, is an important driver of GrIS mass loss next to radiative fluxes (Fausto et al, 2016 ; Kuipers Munneke et al, 2018; van Tiggelen et al, 2020).

Fausto RS, van As D, Box JE, et al (2016) Quantifying the surface energy fluxes in South Greenland during the 2012 high melt episodes using in-situ observations. Front Earth Sci 4:1–9. https://doi.org/10.3389/feart.2016.00082

Smeets PCJP, Kuipers Munneke P, van As D, et al (2018) The K-transect in west Greenland: automatic weather station data (1993–2016). Arctic, Antarct Alp Res 50:. https://doi.org/10.1080/15230430.2017.1420954

Kuipers Munneke P, Smeets CJPP, Reijmer CH, et al (2018) The K-transect on the western Greenland Ice Sheet: Surface energy balance (2003–2016). Arctic, Antarct Alp Res 50:S100003. https://doi.org/10.1080/15230430.2017.1420952

Fausto RS, van As D, Mankoff KD, et al (2021) PROMICE automatic weather station data. Earth Syst Sci Data Discuss 1–41. https://doi.org/https://doi.org/10.5194/essd-2021-80

Van Tiggelen M, Smeets PCJP, Reijmer CH, Van den Broeke MR (2020) A Vertical Propeller Eddy-Covariance Method and Its Application to Long-term Monitoring of Surface Turbulent Fluxes on the Greenland Ice Sheet. Boundary-Layer Meteorol. https://doi.org/10.1007/s10546-020-00536-7

L20: "can be" is poor rationale for studying something. Please revise with something stronger, perhaps relative to radiative heat fluxes.

We propose to modify this sentence (see our reply above).

L22-26: Again, please provide references to backup these statements. A paragraph in the introduction without any references indicates that the research is poorly motivated or that the authors have a complete lack of respect for previous research on this topic. Please revise.

We propose to add several references here to motivate this research further.
L22:

Although the strong vertical temperature gradient provides the required source of energy, it is the persistent katabatic winds that generate the turbulent mixing through wind shear (Forrer & Rotach, 1997; Heinemann 1999). Additionally, the surface of the GrIS close to the ice edge is very rough (Yi et al, 2005, Smeets & Van den Broeke, 2006). It is composed of closely spaced obstacles, such as ice hummocks, crevasses, melt streams and moulins. Due to the effect of form drag (or pressure drag), the magnitude of the turbulent fluxes increases with surface roughness (e.g. Garratt, 1992), thereby enhancing surface melt (Van den Broeke, 1996; Herzfeld et al, 2006). As of today, the effect of form drag on the sensible heat flux over the GrIS, and therefore its impact on surface runoff, remains poorly known.

Garratt, J. R.: The atmospheric boundary layer, Cambridge University Press, Cambridge, 1992.

Forrer J, Rotach MW (1997) On the turbulence structure in the stable boundary layer over the Greenland ice sheet. Boundary-Layer Meteorol 85:111–136. https://doi.org/10.1023/A:1000466827210

Yi, D., Zwally, H. J., and Sun, X.: ICESat measurement of Greenland ice sheet surface slope and roughness, Ann. Glaciol., 42, 83–89, https://doi.org/10.3189/172756405781812691, 2005.

Smeets, C. and Van den Broeke, M. R.: Temporal and spatial variations of the aerodynamic roughness length in the ablation zone of the greenland ice sheet, Boundary-Layer Meteorol., 128, 315–338, https://doi.org/10.1007/s10546-008-9291-0, 2008.

Herzfeld UC, Box JE, Steffen K, et al (2006) A Case Study or the Influence of Snow and Ice Surface Roughness on Melt Energy. Zeitschrift Gletscherkd Glazialgeol 39:1–42

Van den Broeke MR (1996) Characteristics of the lower ablation zone of the West Greenland ice sheet for energy-balance modelling. Ann Glaciol 23:7–13. https://doi.org/10.3189/s0260305500013392

L37: What do you mean by "confined accessible areas"? Please provide some examples.

We refer to areas that are accessible on glaciers for long-term in situ measurements, so not the heavily crevassed areas or very remote areas. We propose the following clarification:

L37:

Historically, the surveying of rough ice was spatially limited to areas accessible for instrument deployment, possibly introducing a bias when it comes to quantifying the overall roughness of a glacier.

L39: Consider replacing "unmanned" with an ungendered term.

We agree with the referee and therefore propose to replace "unmanned areal vehicle" by "uncrewed aerial vehicle".

L39:

The recent development of airborne techniques, such as uncrewed aerial vehicle (UAV) photogrammetry and airborne LiDAR (...)

L40: What do you mean by "limited". Please be more specific.

We mean that airborne methods only cover portions of a glacier or ice sheet. We propose the following clarification:

40:

While these techniques enable the high resolution mapping of roughness obstacles, they often only cover portions of a glacier or ice sheet.

L41: I am not aware of a satellite altimeter that maps the surface roughness of entire glaciers. The ground sampling distance is not small enough. This sentence also makes it sound like UAVs are completely unnecessary. Please revise and be more specific.

Here we do not refer to roughness specifically, but to satellite remote sensing in general.

Concerning mapping the roughness: ICEsat data was used by Yi et al (2005) to map the roughness over the GrIS, and MISR data was used by Nolin & Mar (2019) to map the roughness of Arctic sea ice.

We propose the following clarification at L41.

Yi D, Zwally HJ, Sun X (2005) ICESat measurement of Greenland ice sheet surface slope and roughness. Ann Glaciol 42:83–89. https://doi.org/10.3189/172756405781812691
Nolin AW, Mar E (2019) Arctic sea ice surface roughness estimated from multi-angular reflectance satellite imagery. Remote Sens 11:1–12. https://doi.org/10.3390/rs11010050

L41:

On the other hand, satellite altimetry provides the means cover entire ice sheets, though the horizontal resolution remains a limiting factor when mapping all the obstacles that contribute to form drag.

L42-44: This sentence about sea ice does not fit here in a paragraph about glaciers and ice sheets, please move somewhere else.

Given the very few methods that were developed to map ice surface roughness using satellite data, we believe that mentioning these studies at this point in the

introduction is beneficial. Yet we propose the following modification to avoid further confusion:

L42:

Depending on the type of surface, parameterizations using available satellite products are possible, as presented for Arctic sea-ice by Lüpkes et al. (2013), Petty et al. (2017), and Nolin and Mar (2019).

L99-100: Presumably Fig. 1b could be referenced here?

Added

L99:

At this site, pyramidal ice hummocks with heights between 0.5 m to 1.5 m are superimposed on larger domes 100 of more than 50 m in diameter (see also Fig. 1b).

L145: missing an "of" between transect and AWS.

Added

L145:

"...140 km transect of AWS..."

L226: I thought you just said that this approach did not require interpolation to 1 m profile?

In Eq. (8) we use ATL03 raw photon data to calculate residual photon elevations. The approach that does not require 1-m interpolation is based on ATL06 data. We propose the following modification for clarification:

L224:

When working with the 1-m interpolated profile, we model the standard deviation of the unresolved topography ($\sigma_{sub}$) according to, ...

L252-259: This text would be more useful in the introduction.

We do also mention the issue of bulk model evaluations at L45-48 in the introduction.

L45:

The third and final challenge is the experimental validation of bulk drag models over remote rough ice areas, which either requires in situ eddy-covariance or multi-level wind and temperature measurements.

L260-274: Some more references to Fig. 6 in this paragraph would be useful to the reader.

We agree with the referee and therefore propose several additional reference to Fig. 6. We have also corrected "$\lambda < 0.05$" at line L270.

L260:

The L69 model (Eq.(A5)) overestimates $z_{0m}$ for $\lambda < 0.04$ at this location (Fig. 6, blue line).

(...)

The method by M98 (Eq. (A6)) does account for the displacement height and, while using the same drag coefficient Cd = 0.25, it gives improved results for

$\lambda < 0.04$ (Fig. 6, green line) compared to L69. The same holds for the model by R92 (Fig. 6, red line).
(...) Using $C_d = 0.1$, all three models perform better for $\lambda < 0.04$ but perform poorly for $\lambda < 0.04$ (Fig. 6, dashed lines).

L285: Please clarify what is mean by "satellite backscatter". I presume you are referring to a satellite radar instrument since ICESat-2 does not measure backscatter.
We refer here to the broadening of a backscattered altimeter signals due to surface roughness. We propose the following modification:
L285:
Climate models and satellite altimeter corrections require information about the larger-scale spatial variability of surface (aerodynamic) roughness.

L288: Fig. 6? This figure does not show an elevation profile.
Corrected, we mean Fig. 5.
L288:
The elevation profile from the UAV survey in box A (Fig. 5) was already compared to the overlapping ICESat-2 profiles in Fig. 4a, while $H$, $\lambda$ and $z_{0m}$ are compared in Fig. 7.

Consider swapping Sections 4.1 to 4.2 and Fig. 5 and Fig. 6. I think it would make more logical to move from small to large scale.
We agree with the referee and thus propose to swap sections 4.1 and 4.2.

**4 Results**

4.1 Evaluation of the bulk drag model forced with a UAV DEM
(...)
4.2 Height of the roughness obstacles (H) estimated from ICESat-2
(...)
4.3 Evaluation of ICESat-2 roughness statistics against UAV DEMs

L396-397: It would be useful to briefly state again why Lettau (1969) is not recommended. Some people may only read the abstract and conclusions.
We agree and propose the following addition:
L396:
On the other hand, the use of the model of Lettau (1969) is not recommended over a rough ice surface, as it does not separate the form drag and the skin friction, and neglects both the effects of the displacement height and of inter-obstacle sheltering.

L399-402: I'm not sure I follow this logic. How do you know that ICESat-2 does not capture snow sastrugi or ice hummocks > 1000 m a.s.l. when your UAV surveys are constrained to <600 m a.s.l.?

As explained in L373-377, we have a crude estimate of these heights from fieldwork photographs. We propose the following addition for clarification:
L399:
Obstacles that are small compared to the ICESat-2 footprint diameter of $\approx$ 15 m, such as ice hummocks found above 1000 m elevation in summer, or snow sastrugi expected year-round at even higher locations on the ice sheet from photographic evidence, are not resolved by the ICESat-2 measurements when used in combination with the methods presented in this study.

L405: It's a bit of stretch to say ICESat-2 cannot map z0m above 1000 m when this study presents no UAV surveys above > 1000 m.
We hope that our study proves that ICESat-2 data can be used in the rough-ice areas below 1000-m elevation, given the uncertainties given in the reply above, and in the discussion. In order to convince the reader that the limitations above 1000 m are due to the ICESat-2 data and not to the bulk drag model, we have added a Figure in Appendix A and some explanatory sentences in the discussion:

L475:
Following the steps above, $z_{0m}$ can be estimated for any $H$ and $\lambda$, which is done in Fig. A1. At areas A, B and site S5, $H$ and $\lambda$ are estimated from the UAV surveys and from ICESat-2 data. At site S6, we assume that $H = 0.6 \pm 0.1$ m and $\lambda = 0.045 \pm 0.015$, based on photographs taken during the end of the ablation season. At the highest site S10, we assume that $H = 0.3 \pm 0.2$ m and $\lambda = 0.02 \pm 0.01$, which are typical values for sastrugi (Andreas, 1995).
L378:
Higher up, the ice hummocks become even smaller and the surface eventually becomes snow-covered year-round. Nevertheless, snow sastrugi, known to reach up to 0.5 m height at site S10 from photographic evidence, still contribute to form drag. This results in a maximum observed value of $z_{0m} = 7 \times 10^{-4}$ m at sites S9 and S10 (Fig. 10). Using a rough estimate for both $H$ and $\lambda$ at S6 and S10, based on photographs taken during the end of the ablation season, yields more realistic values for $z_{0m}$ (Fig. A1) than using $H$ and $\lambda$ from the ICESat-2 elevation profiles. Therefore we conclude that the roughness obstacles are not properly resolved at these locations in the ATL03 data using the algorithm presented in this study, even when the correction using the residual photons scatter is applied.

Figure 1. Most of panel (a) is irrelevant, given that data from S9 are not used in this study. It makes it difficult to see how the ICESat-2 tracks intersect the UAV survey grids (A and B). Please consider removing the picture of S9 and providing a zoomed version of the UAV survey grids around the margins of the ice sheet. In the caption please specify if these are the ICESat-2 reference ground tracks or from an actual ICESat-2 beam (e.g. 1r).
We thank the referee for this suggestion. Yet we believe that a perspective

[Figure]

Figure A1: Estimated $z_{0m}$ using the R92 model with parameterized $C_d$ (Appendix A), as function of obstacle height $H$ and frontal area index $\lambda$. The solid squares denote the estimated $H$ and $\lambda$ at three sites using UAV surveys. The dashed squares are first-order guesses based on photographs. See Fig. 1 for the location of each site.

over the whole K-transect is beneficial for the reader interested in the higher elevations. Especially given our reply to the above comment, Fig.1 is helpful to understand why ICESat-2 does not detect any obstacles above 1000 m elevation. Besides, we do show data from S6, S9 and S10 from both in situ measurements and ICESat-2 in Fig. 10.
We have added a reference to Table 2 in the text and in the caption of Fig.1 where the details about each ICESat-2 beam can be found:

L184:
A typical geolocated photon measurement ATL03 (Neumann et al., 2019) can be seen in Fig. 3 for site S5, and in Fig. 4a for area A. Details about which ICESat-2 measurements are compared against the UAV surveys are provided in Table 2.

Figure 2: What is the rationale for these wind directions? Prevailing wind direction from AWS? Please clarify.
These four wind direction are indeed prevailing wind direction, and were chosen to illustrate the variability of the surface topography.
L97:
Four measured elevation profiles, and a high-resolution orthomosaic image are

[Figure]

Figure 1: (a) Map of the K-transect, with the location of the automatic weather stations and mass balance sites indicated by the pink diamonds. The black boxes A and B delineate the areas mapped by UAV photogrammetry. The large black box indicates the area covered in Figs. 5 and 9. The background image was taken by the MSI instrument (ESA, Sentinel-2) on 12-08-2019. Pixel intensity is manually adjusted over the ice sheet for increased contrast. The green solid lines denote the ICESat-2 laser tracks that are compared to the UAV surveys (Table 2). (b) Sites S5 (06 Sep 2019), S6 (06 Sep 2019) and S9 (03 Sep 2019) taken during the yearly maintenance. Note that no data from the the AWS shown at S9 is used in this study. (c) Location of the K-transect on the Greenland ice sheet.

shown in Fig. 2. These were measured on 6 September 2019 at site S5 (67.094° N, 50.069° W, 560 m) in the locally prevailing wind directions, using UAV photogrammetry, of which the details will be given in Sect. 3

Figure 6: There is no reason for such large x and y axis limits on this figure which makes it difficult to determine the correspondence between the SEC and VPEC dots and modeled lines. Please provide a zoomed version of this figure. We agree with the referee and have reduced the extent of the x and y axis of Fig. 6. We also replaced "Observations" by "Estimated from in situ observations" after the feedback of referee #1.

[Figure]

Figure 6: Modelled $z_{0m}$ at site S5 using three different bulk drag models: Lettau (1969, L69, blue lines), Macdonald et al. (1998, M98, green lines), Raupach (1992, R92, red lines) and using two different values for the drag coefficient for form drag: $C_d = 0.25$ (solid lines) and $C_d = 0.1$ (dashed lines). Solid grey symbols are measurements from sonic eddy-covariance (SEC) or vertical propeller eddy-covariance (VPEC). Additional data are from Van Tiggelen et al. (2020, T20). Pink circles are the model results forced with $H$ and $\lambda$ from UAV photogrammetry, using the R92 model and $C_d$ parameterized using Eq. (A2).

---

## Author Comment (AC4)

**Response to referee 3**

**General Comments**

This manuscript addresses retrieval of surface roughness length on ice sheets using ICESat-2 data profiles. Empirically based retrieval of surface roughness length from satellite observations is an enormously important task; the parameter modulates energy fluxes between the atmosphere and the cryosphere, changes in both space and time, and is poorly known. Current methods of retrieval generalize single point measurements to large expanses of the ice sheet; not only do we not have spatially resolved estimates of this parameter, we lack comprehensive understanding of the variance, range, and uncertainty of the parameter. Thus, the present work is extremely timely and important to the community at large. That said, there are several shortcomings with this work; the applicability is limited to a narrow range of surface types and elevations that form a minority of the ice sheet area, the measurements themselves have large uncertainties and are resolved for specific wind directions that do not match prevailing katabatic patterns, and the validation strategy and data are marginally matched to the task. This study is undeniably useful as it forms a basis for future work to build on; the problem under study is a hard task, and incremental progress should be recognized and iterated with new, separate publications that extend to the rest of the ice sheet. In short, this work is worthy of publication following revisionsthere are specific tasks and issues that should be addressed in the revision, and other issues that can be deferred as 'out of scope' and addressed in distinct publications rather than in the current work.

We thank the referee for his time and his comments. In the text below we respond point-by-point and discuss the changes to the work. The referee comments are written in black. Our answer to each comment is written below in blue. The proposed changes to the original manuscript are then highlighted in red.

**Specific Comments**

The spatial and temporal mismatch of the validation data from S5 is the largest issue with the current submission. The S5 site is the only location that ties together all three components of data used in this study– structure from motion DEMs, ICESat-2 tracks, and empirical measurements of surface roughness length by in situ measurements. Although both the 'A' and 'B' structure for motion boxes are bigger, and overlap with ICESat-2 tracks, the lack of weather station data forces S5 to be the primary validation loci, despite the smaller than desired area/fetch coverage. The large temporal gap (September DEM and in situ measurements, March ICESat-2 data) isn't ideal, and needs to be fixed

(preferably), or explained and justified in greater detail. ICESat-2 was operating in September of 2019– why isn't there coincident data provided? Looking at the track crossings, it appears that September 24 was cloudy to the point of signal loss, but this isn't explained...signal from September 12th is stronger and appears to cover and cross over S5, so why wasn't this data used? What is the justification and the trade space between small spatial mismatches vs large temporal mismatches? Why March? Having data coincident in both time and space for the validation is ideal, and a strong case with reasoning and justification needs to be made as to why September data was not used and/or was not tractable for use.

The single reason why we only use a single ICESat-2 measurement at S5 in March 2019 is because this is the only measurement that exactly overpasses the automatic weather station (AWS). This can be visualized in attached Figure R1. Before 1 April 2019, the ATLAS instrument was not pointing at the reference ground track (RGT), but  $\approx 1.5$  km off, over the K-transect. Conveniently this brief mismatch meant that the ICESat-2 data (track 1169, cycle 02, segment 05, beam GT1L) exactly overpassed the AWS S5 in March 2019 within a few meters, taking into account ice velocity ( $\approx 100 \text{ m yr}^{-1}$ ). After 1 April 2019, ICESat-2 was nominally pointing at the RGT again. Unfortunately the closest ground track number 1169 is located 1.5 km West of S5, which prevents a direct comparison between ICESat-2 track 1169 and AWS S5. Yet track 1344 (beam GT1R & GT1L, segment 03) overpassed S5 during on 25 June and 24 Sep 2019, but the signal cannot be retrieved due to clouds. A possibility to have more coincident data would be to move the location of the AWS. However the practical limitations greatly outweigh the scientific added value, due to the crevassed surroundings which considerably limits the amount of areas suited for safe instrument deployment. Besides, we would still not be able to directly compare AWS data to ICESat-2 tracks because of the different wind fetch directions. We propose to add this important piece of information in the revised manuscript.

**L184:**

A typical geolocated photon measurement ATL03 (Neumann et al., 2019) can be seen in Fig. 3 for site S5, and in Fig. 4a for area A. Details about which ICESat-2 measurements are compared against the UAV surveys are provided in Table 2. Not more than one ICESat-2 measurement exactly overlaps each UAV survey. This is mainly due to the presence of clouds and due to changes in laser pointing orientations in other ICESat-2 measurements, but also due to changes in the studied locations due to ice flow.

While having structure for motion, in situ, and ICESat-2 data all be coincident is ideal, the second best approach is paired validation: coincident UAV and in situ data to validate the method, followed by coincident ICESat-2 and in situ data to validate the scaling to the 1D profile. This is especially appealing since the current work already has separate pairing that is discussed with ICESat-2 and UAV data in boxes A and B in addition to pairing of UAV and

Figure R1: ICESat-2 track location with respect to AWS site S5

in situ data at S5; the only pairing not present is between ICESat-2 and in situ tower measurements. Even if data doesn't simultaneously overlap for all three data sets, finding an overlap between ICESat-2 and the S5 station provides the needed coverage for a compelling validation strategy. I'm unclear on if this is possible, or perhaps why it isn't possible since my expertise is more with ICESat-2 than with tower measurements. My impression is that the most of the weather stations such as S5 collect data in dense time series that are continuous save for maintenance or power outages. Is there a reason why there's not coincidence between S5 and ICESat-2, such as lack of co-occurrence that matches the prevailing wind direction? Some of this is addressed explicitly around line 320, but I'm still skeptical; if wind measurements are occurring in dense (i.e., multi-hertz) time series, brief changes from the prevailing wind direction should still occur, even if they are not sustained on the time scale of hours or days. We thank the reviewer for pointing out two distinct issues: (1) the availability of AWS measurements during ICESat-2 overpasses, (2) the absences of wind directions in the available AWS data that match the ICESat-2 ground tracks. Regarding issue (1), we do have year-round flux measurements using the verticalpropeller eddy covariance method (VPEC), that we compare to both the UAV and ICESat-2 modelled  $z_{0m}$  in Figure 8. We have chosen to only use data from 2017 as this has been previously published and discussed in great detail by van Tiggelen et al (A Vertical Propeller Eddy-Covariance Method and Its Application to Long-term Monitoring of Surface Turbulent Fluxes on the Greenland Ice Sheet. Boundary-Layer Meteorol 176, 441–463 (2020). https://doi.org/10.1007/s10546-020-00536-7). We assume that  $z_{0m}$  estimated in March-April 2017 is the same as during March 2019, and thus conclude that the modelled  $z_{0m}$  by ICESat-2 qualitatively agrees with the measurements. The quantitative analysis is not possible because of issue (2).

Regarding issue (2), the reviewer is right: brief changes in wind direction do